# Antigen footprint governs activation of the B cell receptor

Alexey Ferapontov [1,2,6], Marjan Omer [2,3,6], Isabelle Baudrexel [2,4], Jesper Sejrup Nielsen[2,3], Daniel Miotto Dupont [2,3], Kristian Juul-Madsen [1], Philipp Steen [4,5], Alexandra S. Eklund [2,4], Steffen Thiel [1,2], Thomas Vorup-Jensen [1], Ralf Jungmann [2,4,5], Jørgen Kjems[2,3] & Søren Egedal Degn [1,2] ✉

Antigen binding by B cell receptors (BCR) on cognate B cells elicits a response that eventually leads to production of antibodies. However, it is unclear what the distribution of BCRs is on the naïve B cell and how antigen binding triggers the first step in BCR signaling. Using DNA-PAINT super-resolution microscopy, we find that most BCRs are present as monomers, dimers, or loosely associated clusters on resting B cells, with a nearest-neighbor inter-Fab distance of 20–30 nm. We leverage a Holliday junction nanoscaffold to engineer mono-disperse model antigens with precision-controlled affinity and valency, and find that the antigen exerts agonistic effects on the BCR as a function of increasing affinity and avidity. Monovalent macromolecular antigens can activate the BCR at high concentrations, whereas micromolecular antigens cannot, demonstrating that antigen binding does not directly drive activation. Based on this, we propose a BCR activation model determined by the antigen footprint.

The humoral branch of the adaptive immune system plays a central role in combating pathogens through the production of antibodies and immunological memory governed by B cells[1,2]. B cells recognize antigens via their membrane-bound immunoglobulin (mIg), which, in conjunction with two transmembrane proteins Igα and Igβ (CD79A and CD79B, respectively), functions as a B cell receptor (BCR)[3,4]. Antigen binding by the BCR triggers a signaling cascade that ultimately results in B cell proliferation and differentiation into plasma cells, germinal center and memory B cells. The discrete steps in the signaling cascade downstream of BCR activation are well-defined[5], with the first critical step being phosphorylation of the CD79 immunoreceptor tyrosine-based activation motifs (ITAM) by the membrane-associated kinase Lyn and the cytosolic Src-family kinase Syk. Downstream signaling drives activation of the RAS/ERK, JNK and p38 pathways, while $IP_3$ activation causes $Ca^{2+}$ release and NFAT activation. Secondary to initial activation, cytoskeletal rearrangements drive a reorganization of BCRs on the surface, polarizing the cell, and forming a robust signaling complex. However, despite decades of intense scrutiny, it remains unclear exactly how the initial engagement of the BCR by antigen drives the earliest step in the activation of B cells[6–8].

The classical view of receptor activation is that antigen-driven cross-linking of monomeric BCRs drives a localized concentration of ITAMs on the intracellular leaflet of the membrane, which enables scaffolding of kinases and adaptor proteins[9,10]. This model thus proposes that BCRs exist as monomers on the membrane of resting B cells. Evidence derived from Förster resonance energy transfer and direct stochastic optical reconstruction microscopy (dSTORM) has suggested that BCRs spatially reorganize and form large clusters upon binding of multivalent antigens that can cross-link the BCRs to prompt receptor activation[11–13].

[1]Department of Biomedicine, Aarhus University, Aarhus C, Denmark. [2]Center for Cellular Signal Patterns (CellPAT), Aarhus University, Aarhus C, Denmark. [3]Interdisciplinary Nanoscience Center (iNANO), Aarhus University, Aarhus C, Denmark. [4]Max Planck Institute of Biochemistry, Martinsried, Germany. [5]Faculty of Physics and Center for Nanoscience, Ludwig Maximilian University, Munich, Munich, Germany. [6]These authors contributed equally: Alexey Ferapontov, Marjan Omer. ✉e-mail: sdegn@biomed.au.dk

Yet it has also been reported that monovalent antigen is able to activate the BCR, casting doubt upon the validity of the cross-linking model. Using B cells specific for a peptide derived from hen egg lysozyme, Kim et al. found that binding of the BCR by monovalent antigen, but not a Fab fragment targeting the BCR, could induce receptor activation, although the monovalent antigen failed to promote subsequent antigen presentation[14]. Similar findings were presented for transnuclear B cells recognizing a peptide from ovalbumin[15]. Several studies have proposed that differential actions of Src-family kinases and Syk could fine-tune the responsiveness of the BCR to monovalent versus multivalent ligands[8,16,17]. To rationalize such findings, an alternative model for BCR activation, the dissociation-activation model, proposed that under resting conditions BCRs are found in microclusters with their Fab arms closely packed[8]. Ostensibly, in this resting configuration, the ITAMs of CD79 are packed so tightly that it precludes kinase access. The binding of antigen is envisioned to open the Fab arms, in turn dispersing the ITAMs to permit kinase activity[8].

It remains unclear, however, how BCRs would sense the binding of monovalent antigen, given the known flexibility of the hinge region across immunoglobulin isotypes used in BCRs and the lack of structural evidence for allosteric transmission through the heavy chain[18]. In line with this, other studies found no evidence for monovalent antigens as drivers of BCR activation[19,20].

To date, most studies on soluble antigen interactions with the BCR have been based on the use of haptenized protein carriers or peptide antigens. Such antigen preparations have inherent issues precluding definitive control of valency. Direct haptenization of a carrier at any defined molar stoichiometry invariably yields an ensemble of valencies with a Poisson distribution around the defined stoichiometric center. A prevalent problem encountered with peptides is a tendency for subspecies aggregation, depending on hydrophobic patches and the experimental environment, such as pH and the presence of solutes and proteins[21]. Hence, it is very difficult to exclude that a minor fraction of monovalent antigen preparations in fact contains multivalent subspecies.

Here, we take an unbiased approach to address two fundamental questions to understand BCR activation, namely: (1) What is the resting state distribution of BCRs on the B cell surface, and (2) What are the minimal requirements for antigen-driven activation of the BCR? Using DNA points accumulation for imaging in nanoscale topography (DNA-PAINT) we show that most BCRs are present as monomers, dimers, or loosely associated clusters on resting B cells, with a nearest-neighbor inter-Fab distance of approximately 20−30 nm. Leveraging monodisperse, precision-controlled mono- and polyvalent nanoscaffolded antigens we show that antigen binding in itself is insufficient to drive activation, rather, activation requires a minimal antigen size and rigidity. Based on this, we propose a model for BCR activation governed by the antigen footprint. This has fundamental implications for the understanding of the first step in the activation of B cells to produce antibodies in response to infectious microorganisms, allergens, autoantigens and vaccines.

## Results

### BCRs are distributed on the membrane of naïve, resting B cells

To get a more detailed understanding of the starting point for B cell activation, we investigated the BCR distribution on resting B cells using DNA-PAINT super-resolution microscopy[22]. Imaging of the BCR was performed on untouched, naïve murine B lymphocytes, freshly isolated from B1-8hi knock-in mice[23], and fixed in solution, leaving them unperturbed before preservation. Prior to permeabilization and staining, cells were centrifuged on glass channel slides, and both the IgM and IgD BCRs were quantitatively labeled with an anti-mouse kappa light chain nanobody (κLC-Nb), conjugated to a single docking strand[24] (Fig. 1A and Supplementary Fig. 1). For image analysis, we used 2D TIRF images with an imaging depth of approximately 100 nm, and selected either a rectangular or a round region of interest, covering the majority of the lymphocyte touching the surface but excluding the edges (Fig. 1B). BCR clusters were identified by Density-Based Spatial Clustering of Applications with Noise (DBSCAN) which can detect clusters irrespective of their size without the pre-defined input of an estimated number of molecules[25]. In order to assess the number of individual BCRs for each BCR cluster, we made use of the programmable kinetics of DNA-PAINT and performed quantitative PAINT (qPAINT) analysis[26] (Supplementary Fig. 2). Based on the approximation that the influx rate of imager strands remains constant during the time of imaging, DNA-DNA binding kinetics serve as a direct read-out for the number of molecules present. After calibrating the influx rate with single binding sites (SBS), the cluster size in 31 cells was determined and clusters were grouped in monomers (1 molecule), dimers (2 molecules), small islands (3−9 molecules) and large islands (>9 molecules) (Fig. 1C). We found that among all BCR molecules on the B cell surface, about 25% reside in monomers and 24% in dimers. Even though roughly 75% of all 'clusters' detected are monomers and dimers, the largest proportion of all BCRs on the cell surface resides in small island arrangements of 3−9 molecules (37%) (Fig. 1D). Larger islands >9 BCRs are rare and cannot be detected at all in 23% of the cells, probably owing to their random orientation on the glass surface.

To ensure that our κLC-Nb did not underestimate the number of BCRs, we measured the labeling efficiency using a DNA origami with a known number of target sites. This revealed a Fab labeling density of 80%, and among detected BCRs, we observed on average 1.5 nanobodies bound per BCR (Supplementary Fig. 3). We also evaluated whether the DBSCAN analysis of clustering was robust, by simulating a BCR molecule with two nanobodies bound at a 5 nm distance, at the same density as detected for the BCRs on the B cell surface. DBSCAN analysis of the simulated data showed mostly monomeric arrangements (Supplementary Fig. 4).

We subsequently analyzed the small and large islands further by determining the cluster density and found that most BCR clusters contain around one molecule per $1000 \, nm^2$, irrespective of the molecule number per cluster (Fig. 1E). Assuming an even distribution of BCRs within clusters >3 molecules, this corresponds to an inter-Fab average distance of neighboring BCRs of about 20−30 nm. These results make a direct BCR-BCR interaction in islands unlikely, pointing towards external effects, like actin-confinement or membrane architecture, and not direct BCR-BCR-interaction, as the molecular mechanisms behind BCR assemblies. Our observations suggest that there is an equilibrium of isolated BCRs vs. those that are loosely associated in small and large islands.

In addition to cluster analysis, we estimated total BCR numbers on naïve murine B cells by extrapolating the measured BCR density to a spherical B cell surface, using either our experimentally determined cell diameter or an earlier, lower-range literature-derived estimate[27], and taking into account our 80% labeling efficiency, as well as a correction factor to account for membrane ruffles[28]. We obtained an average of 25,000 BCRs per naïve B cell (Fig. 1F).

### Nanoscaffolding of antigen allows affinity and avidity based control of antibody binding

Having established that BCRs are distributed on the B cell surface in the naïve, resting state, we proceeded to the question of the minimal molecular requirements for antigen-driven B cell activation. We leveraged a locked nucleic acid-based nanoscaffold composed of chemically modified RNA strands, forming a defined quaternary complex, which resembles the Holliday junction (HJ) that occurs naturally in connection with homologous recombination. The complex can be recapitulated in vitro using 4 complementary synthetic oligonucleotides, each of which can be conjugated to biomolecules of interest and purified to homogeneity prior to self-assembly.

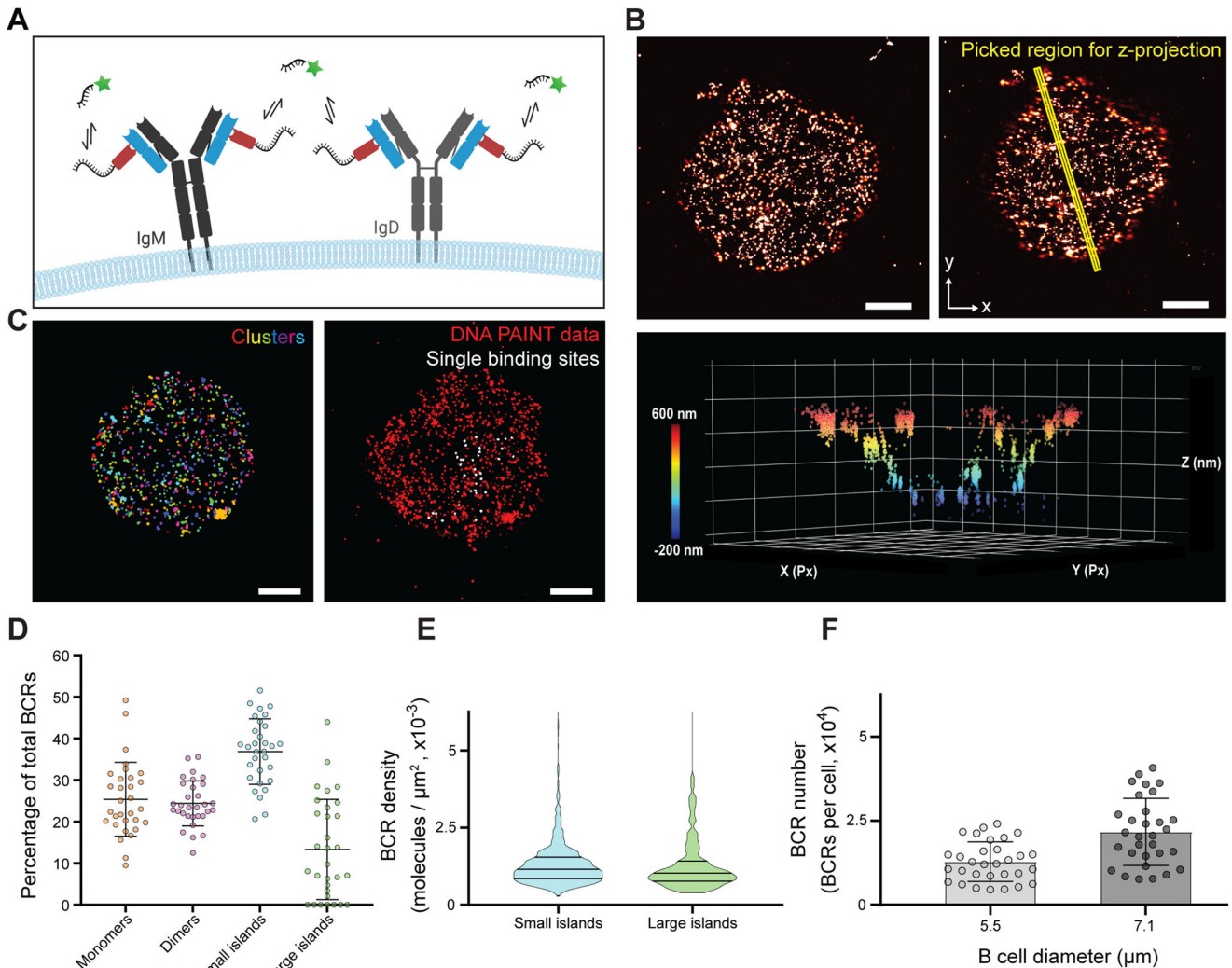

**Fig. 1 | B cell receptor distribution of resting, naïve murine B cells. A** An anti-mouse kappa light chain Nb (κLC-Nb) which binds both IgM and IgD BCRs, conjugated site-specifically to a single docking strand, was used for imaging by the DNA-PAINT method (kappaLc: blue; Nb: red; DNA: black; Cy3b: green). **B** Both 2D (top left panel) and 3D (bottom panel) images of the BCRs were acquired. Plotting a rectangular region in xyz (top right panel) shows a preservation of the round architecture of B cells (bottom panel). Px, pixels. **C** To extract the information on the number of BCRs on the B cell surface along with the surface area, representative regions of the imaged B cells were analyzed by DBSCAN cluster analysis to identify clusters (left panel and Supplementary Fig. 4). For kinetic calibration, single binding sites were picked (right panel, and Supplementary Fig. 2). Imaging data are representative of 3 independent experiments. **D** After calculating the number of BCRs per cluster detected by DBSCAN, the percentage of BCRs found as monomers (1 molecule), dimers (2 molecules), small islands (3–9 molecules) and large islands (>9 molecules) was determined. Bars indicate mean ± SD of a total of $n = 31$ cells from two independent experiments. **E** qPAINT analysis was used to calculate the BCR density on the imaged surface, taking into consideration the labeling efficiency (Supplementary Fig. 3). The lines in the violin plots indicate the quartiles and the median. **F** Total BCR numbers were estimated to be around 15,000 for 5.5 µm B cell diameter (literature[27]) and 25,000 for 7.1 µm diameter (measured). Bars indicate mean ± SD. For images, all scale bars are 1 µm. Source data are provided as a Source Data file.

Incorporation of locked nucleic acids and 2′-OMe RNA nucleotides ensures high thermal stability and resistance to degradation in bio-fluids, enabling cell targeting[29]. Using the nanoscaffold, we precision-engineered monodisperse and stoichiometrically defined model anti-gens with exact antigen valency from 0–3 units. We employed the well-characterized B1-8 system, displaying specificity for the hapten 4-hydroxy-5-nitrophenylacetate (NP) and the heteroclitic hapten 4-hydroxy-3-iodo-5-nitrophenylacetate (NIP)[30]. By combining this with the B1-8i[31] and B1-8hi[23] mouse models of intermediate and high affinity, respectively, we could control antigen affinity across more than 2 orders of magnitude (>400-fold).

Three of the oligos were either left non-functionalized or coupled to NP or NIP, while the fourth oligo was coupled to Alexa Fluor 647 dye (AF647) for subsequent detection, purified by reverse-phase high-pressure liquid chromatography (RP-HPLC), and analyzed by dena-turing PAGE and liquid chromatography–mass spectrometry (LC-MS)

(Fig. 2A and Supplementary Fig. 5). HJs displaying 0–3 copies of hap-tens and an AF647 fluorochrome (Fig. 2B) were prepared by combining oligo conjugates in equimolar ratio in a one-pot self-assembly reaction. Electrophoretic mobility shift assay showed well-defined bands of fully assembled HJs with AF647 fluorochrome incorporation (Fig. 2B, C). The relatively small size of NP and NIP molecules precluded a notice-able shift of the assembled HJ-NP or HJ-NIP constructs compared to the naked HJ.

We employed flow-induced dispersion analysis (FIDA)[32–34] to validate the monodispersity of our hapten-modified HJs (Supplemen-tary Fig. 6B). Following incubation in the presence of serum albumin or all serum proteins, the apparent hydrodynamic radii of HJ-NP or HJ-NIP variants were measured within the interval of $R_h$ ~1.9–2.6 nm (Fig. 2D). Only a minor increase in $R_h$ (~0.6 nm and 0.3 nm for HJ-3xNP and HJ-3xNIP, respectively) was observed, and only for HJs bearing 3 NP or 3 NIP in the presence of serum, presumably due to association with

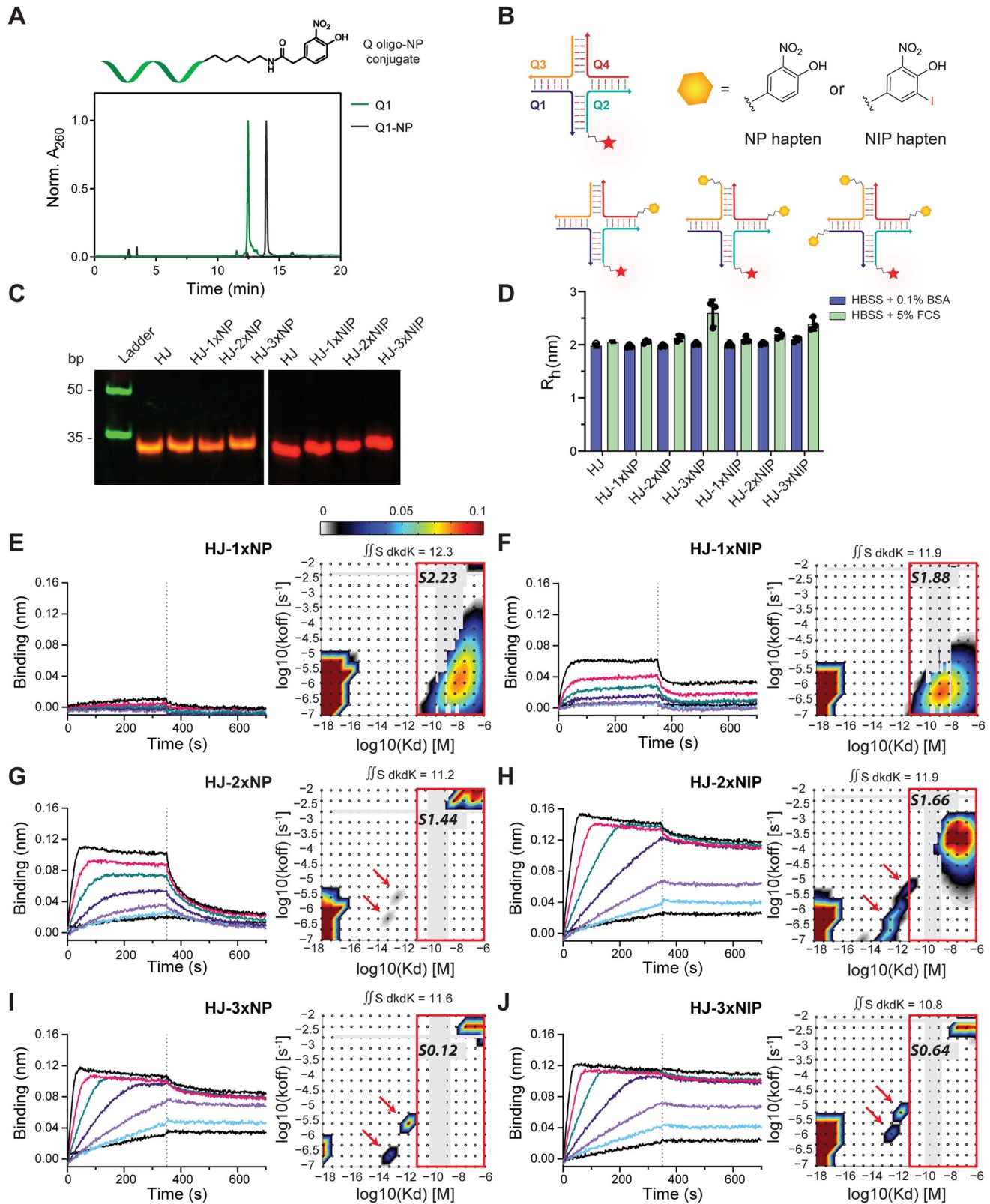

serum proteins. This demonstrated that hapten-modified HJs remained monodisperse under the respective conditions and did not form aggregates or oligomers. In addition, we investigated the in-solution complex formation dynamics of hapten-conjugated HJs at various B1-8i anti-hapten mAb concentrations (Supplementary Fig. 6D). No supramolecular immune complex formation was observed, but we saw antibody association with haptenated HJs

corresponding to hapten valency; e.g., as expected, HJ-3xNIP bound maximally three anti-NP antibodies. Based on the FIDA analysis the $K_D$ for the interaction of anti-NP mAb with HJ-1xNIP was ~20 nM, in line with previously reported affinities[35,36]. Hence, conjugation of hapten to HJ did not alter antibody-antigen binding.

To test the binding capacity of HJ constructs with NP or NIP haptens displayed in various valencies, we performed bio-layer

**Fig. 2 | Assembly of hapten-functionalized HJs and their binding profile to B1-8 anti-NP mAb. A** A representative RP-HPLC chromatogram of Q1 oligo-NP conjugate purification. **B** Schematic representation of the core HJ structure and HJs bearing 1–3 NP or NIP haptens. **C** Native PAGE gel showing the assembly of HJs carrying up to three NP/NIP conjugates and a Q2 oligo bearing AF647 for detection. Green signal is SYBR Gold stain and red signal is AF647. Size marker used is an ultra-low range DNA ladder. **D** Apparent hydrodynamic radii ($R_h$) of Atto488-modified HJs with 0–3 units of NP or NIP (50 nM) in Hanks Balanced Salt Solution (HBSS) supplemented with 0.1% BSA (purple) or 5% FCS (green), determined by FIDA. Bars indicate mean ± SD of $n = 3$ from three independent experiments. **E, G, I** Bio-layer interferometry (BLI) sensorgrams indicating the binding profile of HJ-NP constructs

(0.31–20 nM) to anti-NP mAb (left panels), and modeled distributions of binding kinetics from BLI binding data for haptenated HJ constructs, derived from EVILFIT analysis (right panels). **F, H, J** Binding profile of HJ-NIP constructs (0.22–14 nM) (left panels), and modeled distributions of binding kinetics from BLI binding data for haptenated HJ constructs, derived from EVILFIT analysis (right panels). Distributions are presented as contour plots on 2D grids with $\log_{10}(K_D)$ and $\log_{10}(k_{off})$ on the x- and y-axes, respectively, at signal max 0.1. Binding traces and the best fits are shown in Supplementary Fig. 8. Data in (**E–J**) are representative of at least 3 independent experiments with varying conditions. BSA, bovine serum albumin; FCS, fetal calf serum; NP, 4-hydroxy-3-nitrophenylacetyl; NIP, 4-hydroxy-3-iodo-5-nitro-phenylacetyl. Source data are provided as a Source Data file.

interferometry (BLI). Recombinant B1-8 anti-NP mAb immobilized on protein A biosensors was probed with HJs bearing 0, 1, 2 or 3 copies of NP or NIP in a concentration range of 0.22–20 nM. NP-conjugated ovalbumin (Ova-$NP_{15}$) was included as positive control. Ova-NP showed strong antibody binding with little dissociation, while no binding was detected for the naked HJ scaffold (Supplementary Fig. 7A). HJ constructs bearing NP or NIP displayed specific and enhanced binding interactions in accordance with hapten copy number (Fig. 2E–J). Increasing the valency of hapten resulted in improved binding to anti-NP mAb, particularly for the low-affinity hapten NP, confirming an increased functional affinity (avidity) effect as a result of multivalent interactions.

To obtain a more quantitative evaluation of the avidity effects, we employed EVILFIT analysis[37,38], which calculates the minimal distribution in 1:1 binding kinetics for heterogeneous ligand interactions (Fig. 2E–J and Supplementary Fig. 8), including those produced by multivalent ligands[39]. The 1:1 binding kinetics are typified by the association rate, $k_{on}$, and the dissociation rate, $k_{off}$. For convenience, the plots are made with $K_D$ on the abscissa as calculated from $K_D = k_{on}/k_{off}$, and $k_{off}$ on the ordinate, and the abundance of each type of interaction indicated on a color scale bar. EVILFIT analysis for monovalent NIP and NP HJ data were dominated by a broad distribution of interactions with $K_D$ ~$10^{-9}$–$10^{-8}$ M with matching $k_{off}$ rates in the range of $10^{-7}$–$10^{-3.5}$ s$^{-1}$. For bi- and trivalent NIP and NP HJs, with stronger BLI signals, we observed distributions including interactions, which better matched well-defined monovalent engagements, still with $K_D$s of $10^{-7}$ and $10^{-6}$ M, but with faster $k_{off}$s at ~$10^{-2}$ s$^{-1}$, in line with previous SPR data[35,36]. Importantly, two, mostly discrete, ensembles of interactions in the range of $K_D$ ~$10^{-14}$–$10^{-12}$ M, marked by red arrows, also appeared for these multivalent ligands. Their exponential increase in affinity, differing more than 5 orders of magnitude compared with the monovalent interaction, is a direct estimation of avidity based effects. To provide a robust quantification of the impact of ligand multivalency, we also calculated the sum of low-affinity interactions and their contribution to the extrapolated total signal at binding saturation. Signals were compared at ligand concentrations ranging from 0.32 to 20 nM for NP and 0.22 to 14 nM for NIP HJ. The signal for both NP and NIP HJs ranged from 10.8 to 12.3 Signal units (S) when summed over all ensembles in the plot. The low-affinity contribution was markedly reduced for the multivalent species, decreasing from 2.23S and 1.88S to 0.12S and 0.64S for the NP and NIP HJs, respectively. Taken together, these data showed the ability of the multivalent constructs to form high-avidity interactions as well as shift the interactions quantitatively away from monovalent interactions.

### Nanoscaffolded antigen displays affinity- and avidity-dependent binding to cognate B cells

We next investigated the binding of HJ constructs to cognate B cells using flow cytometry and confocal microscopy. To this end, we employed murine B cells purified from the B1-8hi knock-in BCR model, which carries, on one allele of its immunoglobulin heavy chain locus, the pre-rearranged $V_H D_H J_H$ of the B1-8 clone, that additionally harbors a point mutation increasing baseline affinity 40-fold over the germline

configuration[23]. Although most B cells in this model express the knock-in heavy chain, only a fraction of these express the VL λ1 required for NP-binding, providing a substantial internal negative control population (Supplementary Fig. 9). We estimated the number of total available NP-specific BCRs a priori, based on routinely observing approximately 12% NP-binding B cells in our untouched MACS-purified B cell preparations and relying on our measurement of ~25,000 BCRs per naïve, mature B cell (Fig. 1). HJ constructs containing 0, 1, 2 or 3 haptens were incubated with B cells at a range of different stoichiometries, from 600-fold excess of HJ:BCR down to a 2:1 ratio. Final stoichiometries were confirmed a posteori by the observed NP-specific B cell frequency within each experiment. We observed B cell binding by flow cytometry for all NP and NIP constructs (Fig. 3A), but not for control HJs without hapten (Supplementary Fig. 10A). As we approached 2:1 stoichiometric ratio between HJs and BCRs at the conditions employed, we observed complete disappearance of B cell binding for monovalent NP and NIP HJ, and a shift in median fluorescence intensity (MFI) for the polyvalent HJ positive B cell population. This demonstrated an affinity-, avidity- and concentration-based dependence of haptenized HJ construct binding to BCRs of cognate B cells.

To further validate the binding, we imaged the B cells using confocal microscopy, after exposure to the HJ constructs (Fig. 3B). Naked HJs did not bind B1-8hi B cells, whereas HJs carrying a single hapten yielded a weaker signal compared to polyvalent HJs. The results indicated a statistically significant increase in AF647 signal, which correlated with the number of haptens per HJ, with 3 haptens yielding the highest signal (Fig. 3C, D).

Taken together, our flow cytometric and confocal imaging analyses demonstrated discrete, titratable binding of hapten-conjugated HJ constructs to the B1-8hi subpopulation, correlating with affinity and valency.

### Antigen exerts agonistic effects on the BCR as a function of increasing affinity and avidity

Having observed concentration-dependent and specific binding of monovalent as well as polyvalent hapten-HJ conjugates to B1-8hi B cells, we next asked to what extent this binding induced B cell activation. To this end, we used ratiometric calcium flux analyses based on the calcium-sensitive fluorochromes Fluo-3 and Fura Red. Again, we estimated the relative stoichiometries of antigen to BCR and titrated HJs across a range of stoichiometries from supraphysiological excess (7500:1) to below equimolar stoichiometry (0.75:1) (Fig. 4 and Supplementary Fig. 11). Importantly, calcium flux analyses were carried out under near-equilibrium conditions in the absence of washing steps, and moreover, based on the AF647 signal, we could follow the binding of HJs simultaneously with reading out the calcium flux (Fig. 4A). At high stoichiometries of HJs, both polyvalent and monovalent constructs caused calcium influx, resulting in a change of Fluo-3 to Fura Red signal ratio and indicating cellular activation (Fig. 4B and Supplementary Fig. 11). Furthermore, both types of haptens activated the cells, with NP activation being weaker than NIP as expected from their known affinity difference. When the ratio of HJs to BCRs was lowered to 75:1, the addition of monovalent HJs did not result in B cell activation

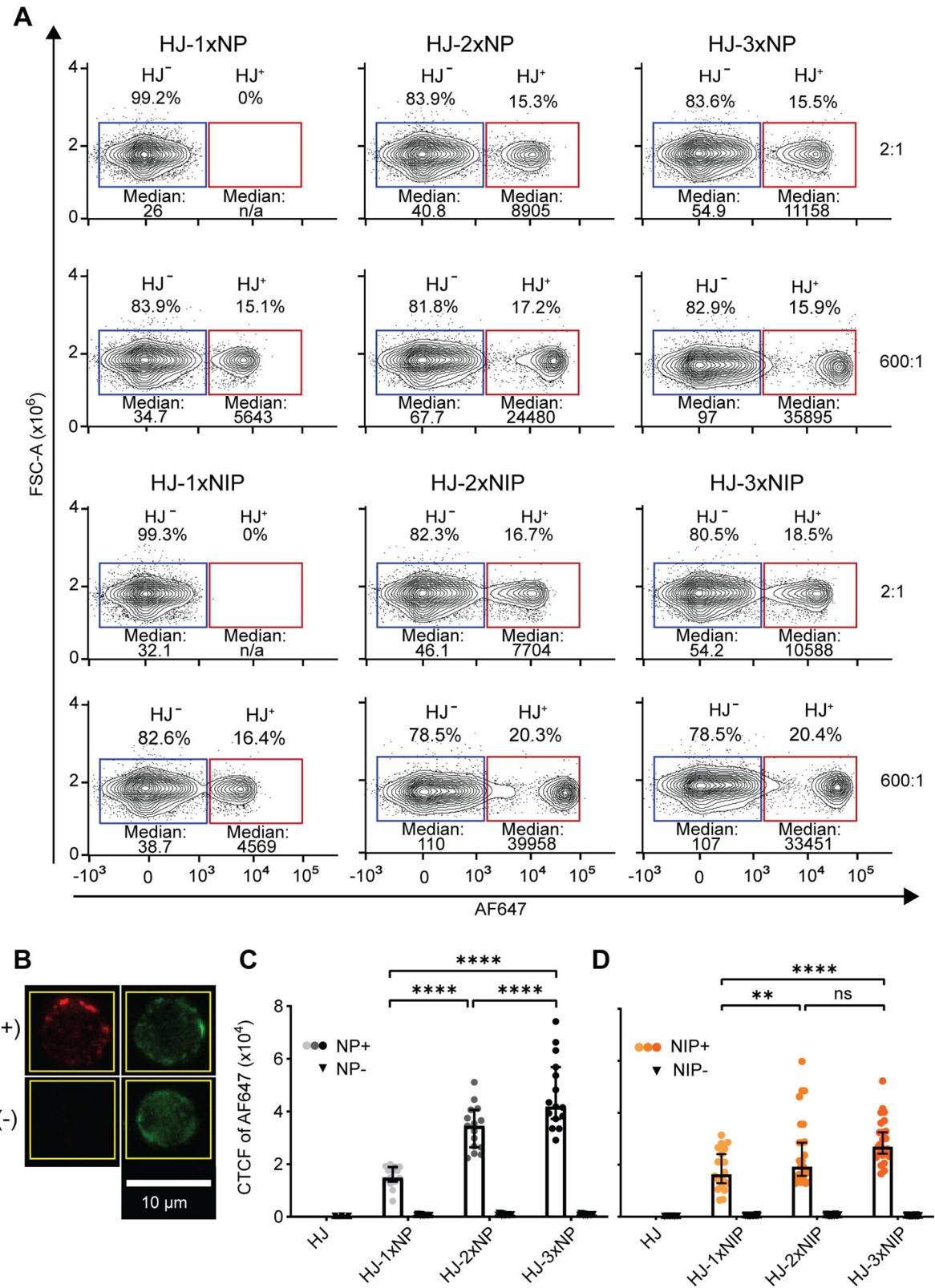

(Fig. 4C). The signal for HJ-1xNIP was below background and no signal was detectable for HJ-1xNP. Importantly, this was seen under conditions where equilibrium calculations still indicated a significant degree of receptor occupancy by HJ-1xNIP, on the order of approximately 19,700 BCRs vs. 900 BCRs, based on 7,500:1 vs. 75:1 stoichiometry of HJ:BCR. Of note, this was based on the specific conditions of our assay, i.e., 31 vs. 0.31 nM HJ in-well to an estimated 4/8 pM BCR/Fab available

based on 200,000 B cells with 25,000 BCRs per cell, 2 Fab per BCR, ~10% NP-specific B cells, and using an affinity constant of B1-8hi for NIP of $K_D = 8.25 \times 10^{-9}$ M[35,36]. The determinant of binding is the relative concentration of antigen to Fab and the $K_D$ of the specific interaction, but for simplicity, we refer to stoichiometry under standard conditions with a fixed number of available Fab arms for each model studied. Moreover, we could experimentally verify 'in-assay' that at the low

**Fig. 3 | Binding analysis of HJ constructs with B1-8hi B cells. A** Purified B cells from B1-8hi mice were analyzed for binding to HJ-0/1/2/3 NP or NIP using flow cytometry. Median Fluorescence Intensity (MFI) is indicated for each gate. Dilutions of HJ-0/1/2/3 NP or NIP were studied, ranging from 600:1 to 2:1 stoichiometry between HJs and BCRs. Data are representative of 2 independent experiments. **B** Representative confocal images of cells that bound both HJ constructs and B220 (Positive, +), or B220 only (Negative, −). The binding of HJs is indicated by AF647 signal (red), whereas the binding of B220 is indicated by green signal (FITC). The images show the same field of view, with one positive and one negative cell.

**C, D** B cells that bound HJ constructs were analyzed based on Corrected Total Cell Fluorescence (CTCF) of AF647 fluorescence signal within the area in the yellow square. Each dot indicates an individual cell (**C**, n = 15 cells per condition, **D**, n = 20 cells for HJ-1xNIP and 25 cells each for HJ-2xNIP and HJ-3xNIP), and bars indicate median values with 95% CI. Imaging data in (**B**) are representative of 3 independent experiments, quantitative data presented in (**C**, **D**) are derived from multiple imaging sessions in one experiment. Statistical analysis was done using two-way ANOVA with Tukey's multiple comparisons test, ns not significant, $p \geq 0.05$; $^{**}p = 0.0049$; $^{****}p < 0.0001$. Source data are provided as Source Data file.

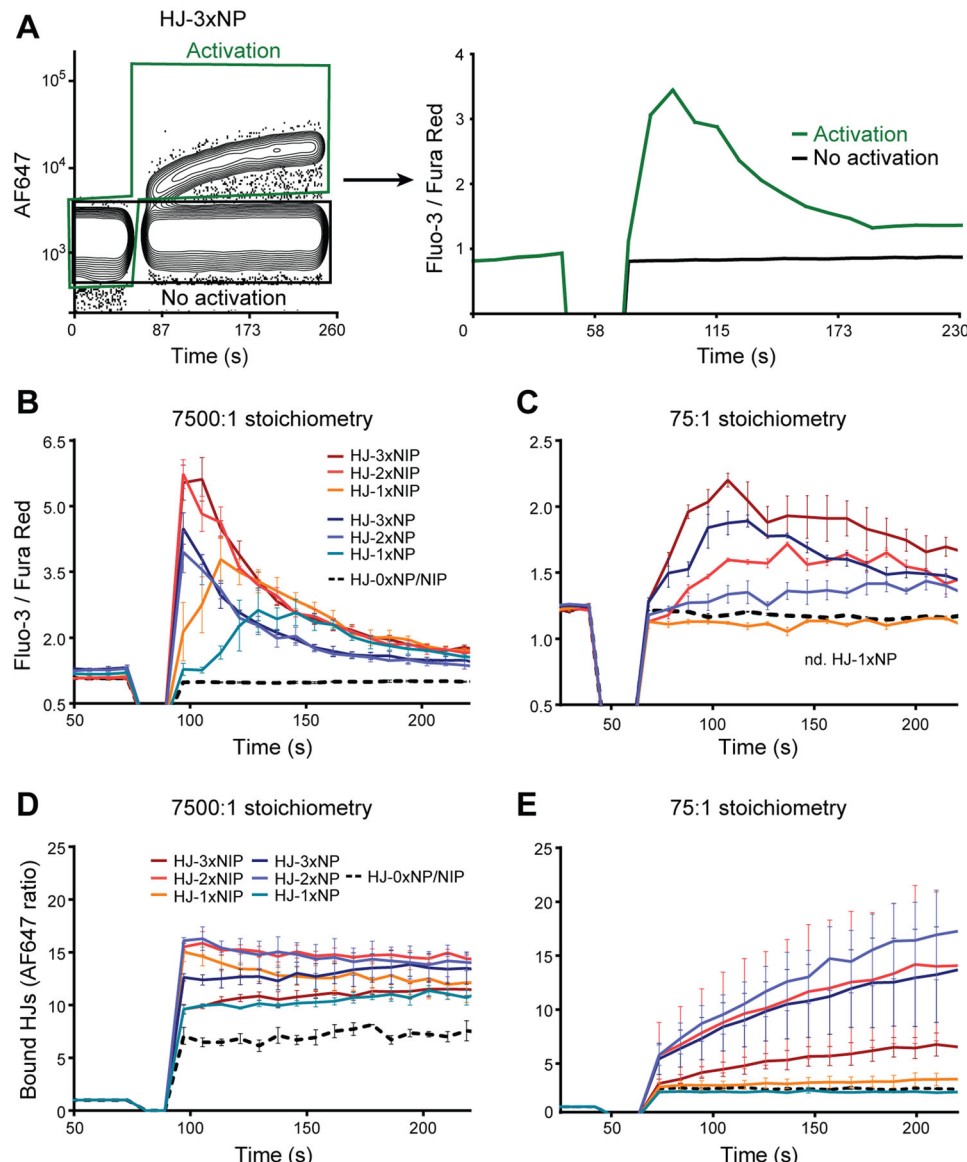

**Fig. 4 | Activation of B1-8hi B cells with HJ constructs. A** B1-8hi B-cell activation was studied in terms of calcium flux by flow cytometry. Left, binding of HJ followed by AF647 signal. Right, calcium flux of cells upon stimulation with HJ-3xNP in the HJ binding (activation) vs. non-binding (no activation, i.e., background) population. **B** HJ constructs with various valencies (0, 1, 2 or 3) of two types of haptens (NP or NIP) at high stoichiometry of HJ constructs to BCR (7,500:1). **C** As (**B**), but at low stoichiometry (75:1). **D** Binding of HJ as followed by AF647 signal for calcium flux graphs shown in (**B**). **E** Binding of HJ as followed by AF647 signal for calcium flux graphs shown in (**C**). In (**B**–**E**), mean ± SD of 3 independent experiments are shown. Source data are provided as Source Data file.

stoichiometry, HJ-1xNIP did indeed bind the B cells, and in general, activation was not a simple function of degree of 'loading' of the cells with HJ (Fig. 4D, E). For example, whereas HJ-3xNIP at high stoichiometry only yielded a binding signal comparable to that of HJ-1xNP (Fig. 4D), it elicited a much more robust activation (Fig. 4B).

Together with our binding studies, our calcium flux analyses suggested that at more physiologically relevant levels, monovalent antigens can be bound by B cells through their BCRs, yet fail to activate the cells. At supraphysiological concentrations of monovalent antigen, however, B cell activation did occur.

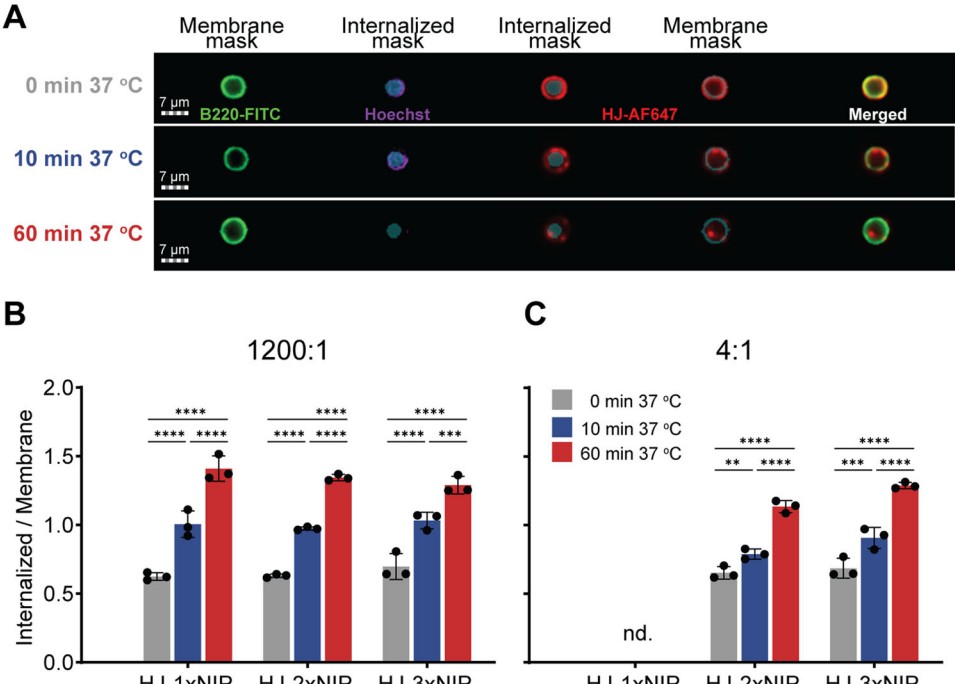

**Fig. 5 | Uptake and internalization of HJs by B1-8hi B cells measured by imaging flow cytometry. A** Representative images from imaging flow cytometry analyses of uptake and internalization of HJs by B1-8hi B cells. A masking method was used to measure the amount of HJ signal (red) on the surface of the cells and within the cells. Membrane and internalization masks were made based on the B220 marker (green), either overlapping with it or excluding it. Nuclear Hoechst stain (blue) shows the potential overlap of internalized mask with the nucleus. **B** AF647 signal from different HJ constructs within these masks was quantified across 3,000-5,000 individual cells and the median ratio of internalized/membrane signal was plotted for 1200:1 HJ:BCR stoichiometry, at the different incubation conditions (time and temperature). The errors indicate mean ± SD. Statistical analysis was conducted by one-way ANOVA using Sidak's method for multiple comparison test ($n = 3$ independent experiments). $****p < 0.0001$; $***p < 0.0009$. **C** The same as in (B) but for 4:1 HJ:BCR stoichiometry. The errors indicate mean ± SD. Statistical analysis was conducted by one-way ANOVA using Sidak's method for multiple comparison test ($n = 3$ independent experiments). $****p < 0.0001$; $***p < 0.0006$; $**p < 0.0086$. Raw distributions for membrane/internalized signals are shown in Supplementary Fig. 12. nd not detected. Source data are provided as Source Data file.

## No qualitative difference between activation induced by mono- and polyvalent antigen

In light of earlier findings that monovalent antigen engagement of the BCR could induce receptor activation, but failed to promote subsequent antigen presentation[14], we sought to qualify the observed activation of B cells that occurred with poly- as well as monovalent HJ at supraphysiological antigen levels. To this end, we interrogated the actin-mediated polarization, capping and receptor-mediated endocytosis that occurs downstream of BCR activation using imaging flow cytometry (Fig. 5 and Supplementary Fig. 12). To generate a baseline for surface bound HJ, we incubated B1-8hi cells with HJ constructs on ice (0 min 37 °C), allowing binding but precluding energy-dependent internalization (Fig. 5A, top). To evaluate the antigen-driven uptake and internalization, cells were incubated at 37 °C for 10 or 60 min after addition of HJ constructs. At 10 min, a clear "capping" phenomenon was observed, where HJs localized in clusters on the cell surface. After 60 min, HJs were increasingly internalized by B cells (Fig. 5). We quantified the ratio between AF647 signal within "membrane" and "internalized" masks to compare how well different HJ constructs were taken up by B cells at low or high stoichiometries. Polyvalent constructs, particularly HJ-3xNIP, were internalized to the greatest degree at both 1200:1 and 4:1 stoichiometries. However, even the monovalent NIP-HJ was internalized to an appreciable extent following incubation at supraphysiological stoichiometry. Although antigen can be passively internalized due to BCR recycling from the membrane, the capping or punctate appearance of antigen and subsequent bulk internalization indicated that there was no fundamental qualitative difference between BCR activation induced by poly- and monovalent HJ antigen.

## Supraphysiological amounts of micromolecular antigens fail to activate B cells

One explanation for the BCR activation occurring at high concentrations of monovalent hapten-HJ constructs could be the appreciable size and potentially the surface charge of the HJs, which could change the microenvironment surrounding engaged BCRs. To investigate this possibility, we examined the ability of micromolecular antigens to induce activation. To be able to assess activation independently of our ability to monitor antigen-binding, we turned to the B1-8i kappa light chain knock-out mouse model (B1-8i Jκ knock-out)[40]. In this model, approximately 50% of the B cells carry receptors specific for NP and NIP, albeit with somewhat lower affinity than B1-8hi cells (approximately 40-fold decreased).

Because B1-8i kappa knock-out mice carry homozygous knock-in of the B1-8i heavy chain, as opposed to the single knock-in of the B1-8hi model and the single recombined heavy chain in wild-type B cells enforced through allelic exclusion, it was possible that they displayed twice the number of BCRs normally present on the B cell surface. Therefore, we examined their surface expression of key cell surface molecules independent of (B220/CD45R), linked with (CD19, signaling component of CD21 co-receptor) or representative of the BCR (CD79 and IgD), compared to either B1-8hi B cells or wild-type C57Bl6/J B cells. B1-8hi cells had slightly lower MFI for IgD than C57Bl/6 and B1-8i cells, which were identical, and no other significant differences were observed (Supplementary Fig. 13). Although flow cytometric evaluation of membrane receptor levels is not absolutely quantitative, the near-identical signals observed indicated a similar level of expression of the BCR irrespective of the knock-in or wild-type provenance of the receptor.

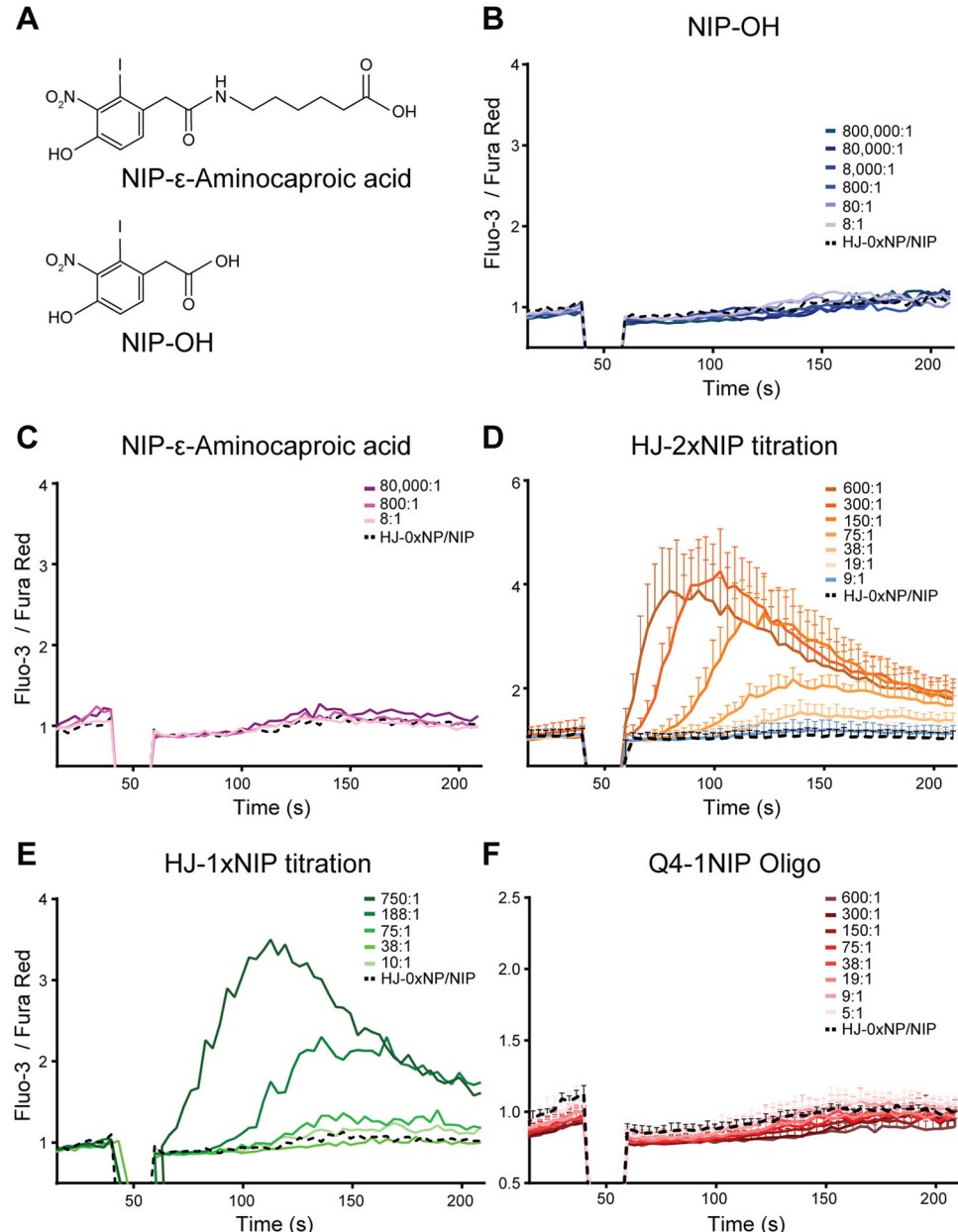

**Fig. 6 | Activation of B1-8i B cells with small-molecule haptens versus HJ constructs. A** Chemical structure of NIP-ε-Aminocaproic acid and NIP-OH. **B** Calcium flux of cells upon stimulation with NIP-OH at varying molar stoichiometries from 800,000-fold molar excess of antigen to BCR, down to 8:1 stoichiometry. **C** Calcium flux of cells upon stimulation with NIP-ε-aminocaproic acid at varying molar stoichiometries from 80,000-fold molar excess down to 8:1 stoichiometry. **D** Calcium flux of cells upon stimulation with HJ-2xNIP at varying molar stoichiometries from 600-fold molar excess of HJ to BCR down to 9:1 stoichiometry. **E** Calcium flux of cells upon stimulation with HJ-1xNIP at varying molar

stoichiometries from 750-fold molar excess of HJ to BCR down to 10:1 stoichiometry. **F** Calcium flux of cells upon stimulation with Q4 oligo conjugated to a single NIP and complexed with its complementary DNA strand (Q4-1NIP), at varying molar stoichiometries from 600-fold molar excess of Q4-1NIP to BCR down to 5:1 stoichiometry. In all panels, background refers to naked HJ stimulation. In (**D**, **F**), mean ± SD of 3 independent experiments are shown, whereas in (**B**, **C**, **E**), representative data from one of two experiments are shown. Source data are provided as Source Data file.

Having established that the level of BCRs displayed by B1-8i cells is in line with that of wild-type B cells, we set up an experiment to interrogate the activation potential of small-molecule hapten antigens. NIP-OH and NIP-caproic acid (Fig. 6A) were titrated onto B1-8i cells and we measured the calcium flux potential. This revealed that even at 800,000-fold molar excess, these antigens were unable to induce activation (Fig. 6B, C). Even considering the lower affinity of B1-8i B cells, these conditions should cause ~98% BCR occupancy, corresponding to ~24,500 engaged BCRs on the cell surface of cognate B cells at the conditions employed. This was based on 16 μM NIP hapten,

20/40 pM BCR/Fab available based on 200,000 B cells with an estimated 25,000 BCRs per cell, 2 Fab per BCR, an observed 50% NP-specific B cells, and using an affinity constant of B1-8i for NIP of $K_D = 3.3 \times 10^{-7}$ M. Importantly, these results, obtained under conditions where a high fraction of BCRs would be occupied, indicated that antigen *binding*, in and of itself, was insufficient to drive BCR activation. To further corroborate this, we employed NP-FITC as small-molecule antigen, in order to be able to directly read-out binding simultaneously with calcium flux. Because of the spectral overlap of FITC and Fluo-3, we had to rely on Fura Red signal alone for calcium

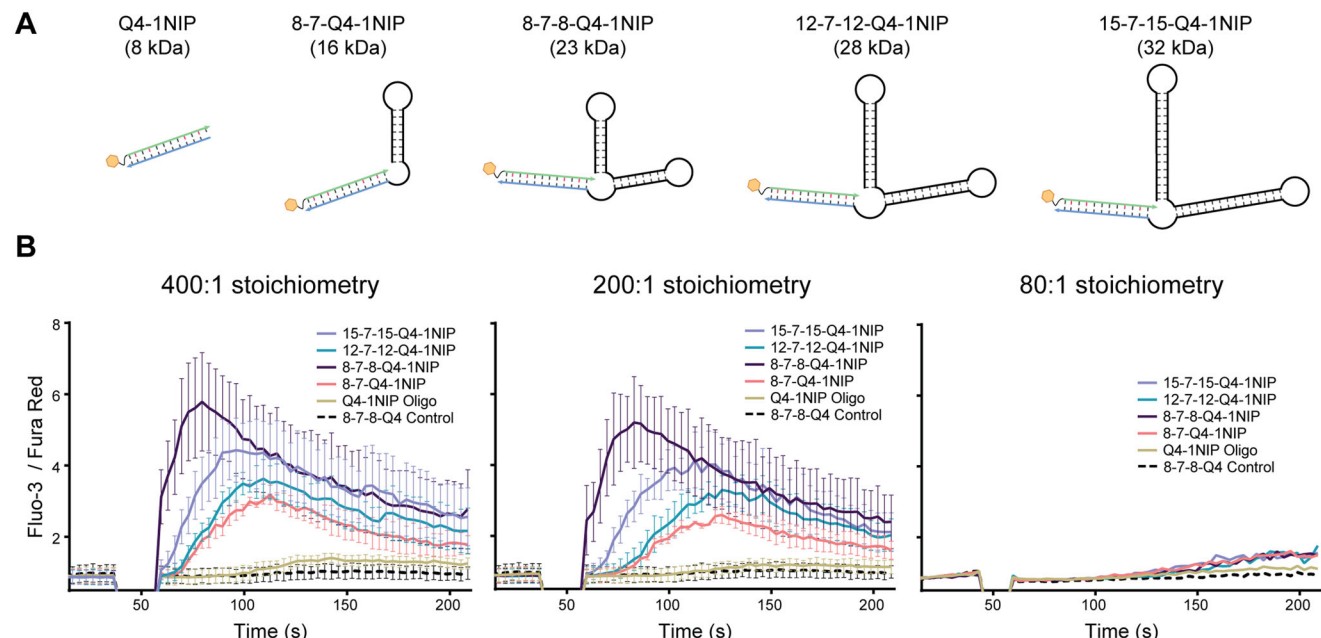

**Fig. 7 | Activation of B1-8i B cells with dumbbell-Q4-1NIP constructs versus Q4-1NIP alone. A** Structures of the 4 dumbbell structure variants assembled with Q4-1NIP. **B** Calcium flux of B1-8i cells upon stimulation with the 4 dumbbell constructs, Q4-1NIP alone, or a control dumbbell structure without NIP, at 400-fold (left), 200-fold (middle) or 80-fold (right) molar excess of antigen to BCR. In B, data represent mean ± SD of 3 independent experiments for 400:1 and 200:1 stoichiometry, and one representative experiment for 80:1 stoichiometry. Source data are provided as Source Data file.

flux measurement. Despite demonstrable robust binding of NP-FITC at 800,000-fold molar excess, this antigen failed to induce activation, confirming that antigen *binding*, in and of itself, was insufficient to drive BCR activation (Supplementary Fig. 14).

We next considered the possibility that the observed difference between monovalent hapten-HJ conjugates and isolated small-molecular weight haptens could be explained by a difference in propensity to aggregation. Despite the documented mono-dispersity of HJ scaffolds, and our data demonstrating that they do not aggregate even in complex environments (serum), it is virtually impossible to exclude that minute levels of aggregates could be formed in situ. To evaluate the likeliness of aggregate formation explaining the observed activation effects, we titrated HJ-2xNIP and determined the minimal stoichiometry that could induce activation and arrived at a ratio around 19:1 HJ:BCR (Fig. 6D). We then determined the minimal stoichiometry for HJ-1xNIP that could induce activation and arrived at around 75:1. Hence, the 1x conjugate was ~4-fold less potent in inducing activation (Fig. 6E). If the activity of the 1x conjugate should be explained by aggregate formation, this would require aggregates equivalent of ~25% 2x conjugate, something which could be ruled out by our FIDA analyses (Fig. 2 and Supplementary Fig. 6). Taken together, this demonstrated that aggregate formation in situ is not the driving force behind the observed activation by monovalent antigen.

To further delineate the minimal requirement for monovalent antigen to exert an agonistic effect on the BCR, we generated a double-stranded variant of the Q4 oligo component of the HJ, by complexing the Q4 oligo conjugated with a single NIP to its complementary DNA strand. Importantly, this construct would have half the mass of the HJ nanoscaffold, ~8 kDa, and carry a single NIP. Yet, due to the extended linear conformation of the double-helix, as opposed to the cruciform shape of the HJ, combined with the rotational freedom around the hapten linkage, we expect this nanoscaffold to occupy a minimal footprint around the Fab binding site, insufficient to drive activation. Congruent with this notion, we did not observe any activation with the Q4-1NIP at ratios severalfold exceeding the minimal activating level of HJ-1xNIP (Fig. 6F).

To further investigate the footprint requirements for activation, we synthesized a range of DNA dumbbell structures, which could self-assemble with the Q4 oligo (Supplementary Fig. 15). This allowed us to generate antigenic structures varying in size from 16 kDa to 32 kDa (Fig. 7A), albeit with a concomitant increase in flexibility. We compared the ability of these Q4-1NIP dumbbell structures to induce activation, using Q4-1NIP alone and Q4 dumbbells devoid of NIP as controls. Whereas Q4-1NIP alone, as previously observed (Fig. 6F), could not activate cells at any of the concentrations employed, all dumbbell structures complexed with Q4 carrying NIP were capable of activating cells at high stoichiometries (400:1 and 200:1) but not at low stoichiometry (80:1) (Fig. 7B). Surprisingly, there was not a linear relationship between the size of the scaffold and its activity. The 8-7-8-Q4-1NIP dumbbell structure elicited the most potent response, suggesting that differences in the flexibility or preferred 3D configuration of the various dumbbells may also influence their activity. Our overall findings regarding affinity- and avidity-dependent activation and the role of antigen footprint in monovalent antigen-driven activation are illustrated schematically in Fig. 8.

## Discussion

The BCR is, arguably, the most versatile cellular receptor: it is generated by somatic recombination to potentially recognize any ligand of virtually any origin, even ligands that never existed in nature, and it is malleable through somatic hypermutation. This versatility and diversity are likely the main reasons why it has remained a significant challenge to firmly define the molecular mechanism of BCR activation. Our findings here address two fundamental questions, as set out in the introduction: (1) What is the resting state distribution of BCRs on the B cell surface, and (2) What are the minimal requirements for antigen-driven activation of the BCR?

To address the first question, we employed DNA-PAINT. Of note, previous attempts to determine BCR distributions on the cell surface have been made, however, classical super-resolution approaches are prone to quantification artifacts such as under- or overcounting due to the unpredictable photophysical blinking properties of fluorescent proteins or organic dyes[41,42]. Prior studies employing PALM and (d)

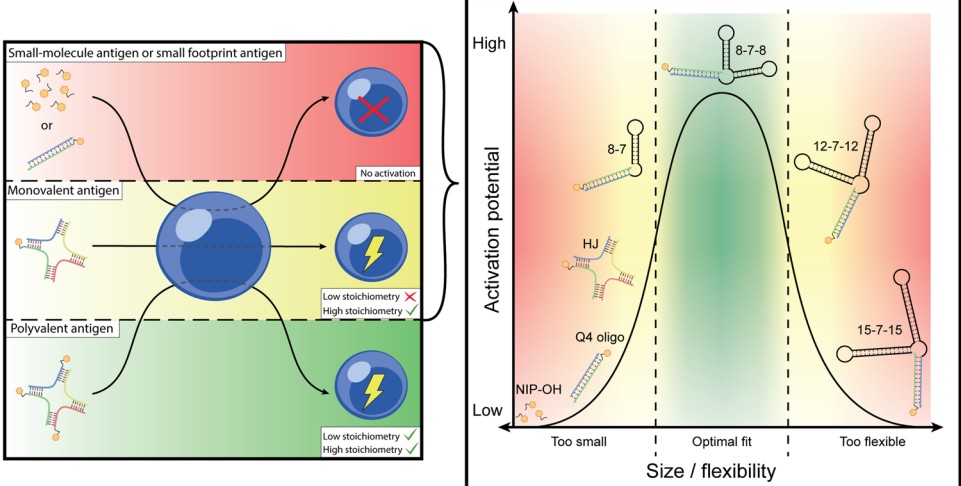

**Fig. 8 | Schematic representation of central findings.** Left: Polyvalent antigen induces robust activation at both low and high stoichiometry, whereas monovalent antigen is only active at high stoichiometry, and small-molecule or small footprint monovalent antigen has no activity whatsoever. Right: The role of footprint in determining monovalent antigen activity. There is not a simple linear relationship between overall size and capacity for activation, rather, a combination of size and rigidity determines the degree of activation.

STORM to provide plasma membrane maps of the BCR and associated components more or less invariably reported nanoscale clusters or larger protein islands[13,28,43,44]. However, similar observations were made for virtually any membrane protein[45], and it has emerged that multiple observations of single fluorophores also lead to clustered localizations that may closely resemble clustered molecules. In turn, this impedes direct inference of the presence of protein clusters based on the observed localization clusters. We circumvented this obstacle by employing DNA-PAINT, which is able to overcome these limitations. Our data indicate that BCRs are not significantly pre-clustered in an organized fashion, as has otherwise been supposed in the dissociation-activation model[46]. Our findings regarding the BCR distribution are, in essence, a corollary to the observation that apparent T cell receptor (TCR) nanoclustering could be attributed to overcounting artifacts inherent to single-molecule-localization microscopy and that both resting naïve and antigen-experienced T cells display monomeric TCR distributions[47,48]. These findings are in line with another recent study on the organization of the BCR using STED microscopy on human B cells[49]. Based on our DNA-PAINT analyses, we estimated that there was an average of 25,000 BCRs per naïve B cell, which is somewhat below the frequently stated estimate of ~100,000 receptors per cell in the literature[50]. However, experimentally determined values for BCR number on naïve, murine B cells range widely, from 12,000 based on dSTORM[44], (total number calculated by us based on reported BCR density of 80 BCR/μm² given in the supplement, with same diameter and correction for membrane ruffles as for our own number) to 335,000 based on flow cytometric quantification[28], (sum of IgD and IgM). Here, we were able to accurately assess our labeling density, and we consider our estimate based on direct single cell measurements to be robust. Accordingly, we employed this number in our subsequent equilibrium calculations and estimates of HJ:receptor ratios.

To address the second question, we employed a nanoscaffold allowing precise stoichiometric assemblies of defined antigenic valency, in a setting that allows precise determination of antigen:BCR ratios. We found that both monovalent and polyvalent configurations could induce activation, although monovalent antigen only exerted an activity at what are likely supraphysiological conditions. When activation occurred, there was no apparent qualitative difference between the activation elicited by mono- and polyvalent antigens. For comparison with our nanoscaffolded antigens, we probed the ability of simple, small-molecule haptens to induce activation, and found that

they failed to do so, even at supersaturating conditions, and despite confirmed binding. Two central points can be drawn from our observations: firstly, it is not the *binding* itself, which induces activation; and secondly, at certain conditions monovalent antigens *are* able to induce activation, albeit inefficiently. Our results reconcile prior conflicting data in the literature, where monovalent antigen has variably been assigned either activating[14,15,51] or non-activating[19,20] capacity, as condition-dependent observations that could, in isolation, support either claim. At the same time, our findings upend this discussion of antigenic activity by the observation that not only affinity and valency, but also additional molecular properties such as size are critical in determining their capacity for BCR activation. We confirmed this experimentally by varying the 'footprint' of a defined antigen using a series of synthetic DNA dumbbell structures.

The activity of larger monovalent antigens is highly concentration-dependent, and robust activation requires what is arguably supraphysiological levels, although on a side note, it is not really known what the antigen availability is in vivo, and what range of relative ratios of antigen to cognate BCRs is encountered in the physiological setting. However, based on common immunization doses of 100 ng–1 μg, with model antigens such as haptenized HEL or Ova (~15 and 45 kDa, respectively), with subcutaneous immunization routes draining to the popliteal or inguinal lymph nodes (containing 1.5–3 million lymphocytes, of which ~40% are B cells, and 1–10% could be antigen-specific), we arrive at antigen to BCR ratios ranging from approximately 450:1 to 270,000:1. Although numerous factors could modify the actual antigen availability in vivo, these considerations substantiate the likely physiological relevance of our experimental conditions. In our in vitro setting, monovalent antigen activity required molar stoichiometries that indicated engagement of a significant number of BCRs at the conditions employed. Notably, in the experiments employing B1-8hi cells, we did not see activation with HJ-1xNIP at 75:1 stoichiometry, whereas we did in the case of B1-8i cells. The difference of these two scenarios is ~10% vs. ~50% binders, and $K_D$'s of $8.25 \times 10^{-9}$ M and $3.3 \times 10^{-7}$ M for B1-8hi and B1-8i, respectively. Equilibrium calculations reveal engagement of ~900 and ~120 BCRs, respectively, for these two scenarios. In light of this, the lack of observable activation in the case of B1-8hi cells is somewhat surprising, and likely an artifact caused by our inability to accurately identify the small (~10%) subpopulation that is fluxing calcium, within the sizeable non-fluxing background. The B1-8i results, however, demonstrate that

monovalent antigen *can* induce activation, and notably can do so under conditions where only a fraction (<1%) of BCRs are engaged. This result is unlikely to be a consequence of in situ aggregation, because the activity of our monovalent antigen preparation was comparable to a 4 times more diluted divalent antigen preparation. Hence, if the activity of the monovalent antigen preparation should be solely explained by a contaminant fraction of aggregate, this should make up at least 25% of the material, and this was easily ruled out by our FIDA analysis. In further support of this, we did not observe any multivalency effects in our BLI and EvilFit analyses of the monovalent antigen preparations.

The 75:1 scenario for NIP-HJ and B1-8i cells also hints that it is unlikely that the activity depends significantly on individual receptor occupancy time, because the relatively low affinity and the low number of BCRs engaged at equilibrium would indicate a continuous unbinding and binding of the monovalent antigen. This notion, combined with the size requirement and demonstration that binding in and of itself is insufficient, suggests that occupation of the BCR with a sizeable antigenic binder is intimately linked with activation.

These findings portray a steric aspect of binding large antigens as central in driving BCR signaling, which also gives new insight on the role of polyvalency for at least some antigens in BCR signaling. The binding kinetic studies clearly revealed that our polyvalent constructs generated an interaction pattern characteristic for other polyvalent molecules, with multiple, discrete ensembles of interactions in spite of their structural homogeneity[52]. This confirms that polyvalent interactions are permitted by the constructs, but comparison with the monovalent antigen excludes that structural properties of the polyvalent interaction, namely clustering, per se, is critical for the increased signaling. In their thermodynamic model of polyvalent interaction, Kitov and Bundle used a step-wise description with one initial intermolecular interaction, followed by multiple intramolecular interactions[53]. The latter would form more easily, once the polyvalent construct has made its initial transition from a soluble to a surface tethered form. In this way, the polyvalency of the constructs can drive additional BCR interactions to a level, which can be mimicked by a monovalent construct at a much higher concentration. Yet in both cases, the critical event for signaling is the engagement between a single antigen and BCR. Our finding that soluble monovalent antigen *can* activate the BCR argues strongly against the simple cross-linking model found in most textbooks, as this allows for only multivalent activities.

The notion that bulky soluble, or membrane-bound monovalent antigen affords activation, whereas micromolecular antigen binding does not, bears a striking resemblance to the scenario of T cell activation, which invariably occurs on membrane surfaces. Although the classical model for T cell activation is also based on clustering[54], an alternative hypothesis is embodied in the kinetic segregation model[55]. The kinetic segregation model posits that under resting conditions, the kinase activity on CD79 ITAMs is counterbalanced by the highly prevalent phosphatase CD45. However, TCR binding to cognate peptide:MHC on an opposing cell surface entails a close approximation of membranes due to the short-range TCR – peptide:MHC interactions. This excludes the extended and bulky CD45 molecule from the nascent synapse and shifts the local balance in favor of kinase activity, thereby initiating the signaling cascade. In its current formulation as a model of T cell activation, the kinetic segregation model requires a juxtaposing cell membrane to exclude CD45 from the signaling complex. This mechanism could easily be extrapolated to BCR sensing of membrane-scaffolded antigen, e.g., on follicular dendritic cells. However, our data are more congruent with something occurring in a much smaller environment, and the model could potentially also be applicable to monovalent and multivalent soluble antigens on the condition that these were bulky enough, or had other properties that allowed them to occlude CD45 from the vicinity of the BCR. In an evolutionary perspective, a common cell biological principle could well govern the mechanism of activation of the TCR and the BCR, despite the inherent differences in the breadth and nature of antigens recognized.

Thus, taken together, our data point to an alternative mechanism of antigen-driven activation through the BCR. Based on the speed with which activation occurs, and the overall congruence of our observations with the kinetic segregation model of T cell activation, we suggest that the B cells are poised for activation in a kinetic equilibrium through the counterbalanced activities of kinases and phosphatases, allowing a relatively small perturbation to rapidly shift the balance in favor of activation. The critical event for signaling is the engagement between a single antigen and BCR. However, the lack of activation by micromolecular antigens, under conditions where a high fraction of BCRs are occupied, demonstrates that antigen *binding*, in and of itself, is insufficient to drive activation. Our finding that a monovalent ~16 kDa antigen can induce activation, while a monovalent ~8 kDa antigen cannot, demonstrates that the antigen needs to occupy a certain footprint around the Fab binding site to trigger activation. This is congruent with our DNA-PAINT imaging results, supporting that the BCR microenvironment rather than physical BCR interactions are responsible for regulating its nanoscale organization. Our further definition of the antigen footprint using DNA dumbbell nanostructures confirm size as a critical parameter, but reveal that rigidity likely plays an additional role, in line with the suggested requirement for the antigen to displace CD45 from the vicinity of the BCR in order to elicit activation. We envision that docking a bulky and rigid structure to BCR(s) on the surface of the B cell will perturb the dynamic equilibrium movement of CD45 into the contact zone, causing a spatial depletion of CD45 from the vicinity of the engaged BCR(s), thereby shifting the balance of kinases and phosphatases in favor of the former. Antigen factors favoring activation would include multivalency (engagement of multiple BCRs), size (larger exclusion zone), and potentially charge (negative charge could increase repulsion of CD45), whereas increased flexibility would disfavor activation (reduction of the exclusion zone).

Our revised model for BCR activation accommodates both monovalent and multivalent antigen activities. Furthermore, it explains qualitative differences between the two in terms of the thermodynamic model of polyvalent interaction rather than simple clustering. This enables a more profound understanding of the complex activities of soluble antigens in their native form and subsequent to antigen scaffolding on membranes. Soluble antigen recognition is arguably the simplest scenario of B cell activation, and that which sets it apart from T cell recognition of peptide MHC; yet in vivo, the BCR activation mechanism must at the same time accommodate antigen scaffolding on membranes, such as the surface of follicular dendritic cells[20,56,57]. The notion that size and valency are important determinants of antigenic activity is additionally well in line with the evolutionary drive to recognize larger particulate material, such as virus particles or bacteria, and to be ignorant of smaller, likely innocuous antigens. It is also congruent with the long-standing notion that haptens do not elicit immune responses in and of themselves. The higher potential of repeated ligands can be rationalized in terms of the common properties of biologically relevant antigens: sizeable particles with repeated structures, such as the repeated carbohydrate structures of the glycocalyx or spike proteins of the virion. Yet the capability to respond to sizeable monovalent antigen averts vulnerabilities in the B cell defense.

## Methods
### Mice
The B1-8hi line (B6.129P2-*Ptprc^a Igh^{tm1Mnz}*/J, Jax strain no.: 007594)[23] was kindly provided by Thomas Winkler, and the B1-8i line (B6.129P2(C)-*Igh^{tm2Cgn}*/J, Jax strain no.: 012642)[31] was kindly provided by Anja Hauser. C57Bl/6JRj mice were purchased from Janvier Labs. The mice were maintained in our SPF vivarium at the Department of Biomedicine, Aarhus University, in IVCs on a standard 12-h light/dark cycle, with

standard chow and water ad libitum, at ambient room temperature (20–22 °C) and ambient humidity. Breeding and maintenance of transgenic animals and harvesting of cells for use in the described experiments were conducted in accordance with the guidelines of the European Community and was approved by the Danish Animal Experiments Inspectorate (protocol number 2017-15-0201-01319).

### Magnetic-activated cell sorting (MACS)

MACS cell separation was performed using a modified version of official Miltenyi Biotec MACS protocol. Mice were euthanized by cervical dislocation, spleens were harvested and processed through 70 μm cell strainers using MACS buffer (Dulbecco's Phosphate Buffered Saline (DPBS), 2% Heat Inactivated Fetal Bovine Serum (FBS), 2 mM EDTA) followed by RBC Lysis. Splenocytes were recovered by centrifugation at 200 g and filtered again through 70 μm cell strainer and obtained cells were treated with Fc Block (Purified Rat Anti-Mouse CD16/CD32, Clone 2.4G2, BD Catalog No.: 553142, Lot. 1293770) for 5 min, followed by Miltenyi Biotec Biotin Ab-cocktail for 30 min. Following that, cells were centrifuged at 200 × g, 4 °C, 10 min, resuspended in 1 mL MACS buffer and mixed with Miltenyi Biotec anti-biotin magnetic beads for 20 min. Cell suspension was then applied to Miltenyi Biotec LS column through 70 μm cell strainer, after which the column was washed with additional 3 mL of MACS buffer. Both flow through (untouched B cells) and column eluate (remaining cells) were collected and analyzed using flow cytometry.

### Primary nanobody structure

The amino acid sequence for the anti-mouse kappa light chain nanobody (κLC-Nb) presented below was kindly provided by Dirk Görlich[24]. The complementarity determining regions (CDRs) are shown in red. For DNA-PAINT super-resolution microscopy, the serine residue at position 87 (Ser87) was replaced with a cysteine residue (Cys87) for site-specific labeling of a DNA oligonucleotide. This design was based on the study by Hansen et al.[58], who obtained high conjugation efficiency with engineered cysteine residues slightly distant from the C-terminus.

5 mM β-mercaptoethanol (BME) prior to sonication for 5 min at 8 watts (20% amplitude) in pulses of 10 s and 10 s of rest. The cell lysate was centrifuged twice at 8000 × g for 20 min before the cleared supernatant containing the Nb was collected and sterile filtered (first using a 0.45 μm filter followed by a 0.22 μm filter).

The Nb was purified using Äkta Start system (Cytiva) by loading the supernatant onto a HisTrap column (Cytiva) packed with Ni Sepharose High Performance affinity resin. Prior to elution, the column was washed thoroughly with binding buffer (PBS, 400 mM NaCl, 20 mM imidazole, pH 7.4, 5 mM BME) to remove unspecific proteins. The Nb were eluted by running a linear gradient of binding buffer with elution buffer (PBS, 400 mM, 500 mM imidazole, pH 7.4). Fractions containing the Nb were collected, and using Amicon Ultra-4 Centrifugal Filter Units (3 K cutoff), the proteins were buffer-exchanged and concentrated in PBS following the manufacturers' instructions.

### Nanobody conjugation of DNA-PAINT oligonucleotide

Initially, the κLC-Nb was reduced with 5 mM tris(2-carboxyethyl) phosphine (TCEP) (ThermoFisher) for 2 h on ice. Subsequently, the Nb was conjugated to the DNA-PAINT oligonucleotide 7xR3 (Table S1) in a two-step reaction. First, the C-terminus cysteine was reacted with 20-fold molar excess of the heterobifunctional linker DBCO-PEG4-maleimide O/N at 4 °C in a thiol-maleimide reaction. Excess linker was removed using Zeba Spin Desalting columns (7 K MWCO), and the sample was buffer-exchanged to PBS containing 1 mM EDTA, pH 8. Subsequently, the DBCO-modified nanobody was reacted with 5'-azide-modified DNA oligo in 1:5 molar ratio and incubated at 4 °C O/N. The final nanobody-DNA conjugate was purified by anion exchange chromatography on a Source Q 5/50 column (GE Healthcare) using a linear gradient (1 mL/min) with PBS, pH 7.4 as binding buffer and PBS + 1 M NaCl, pH 7.4 as elution buffer.

### Antibody conjugation of DNA-PAINT oligonucleotide

The mouse IgG2a (mIgG2a) antibody (BALB/c IgG2a,κ Purified Mouse Anti-Mouse I-A[b], Clone AF6-120.1, BD Pharmingen, Catalog No.:

```
         10              20              30              40              50              60
MGQVQLVESG      GGWVQPGGSL      RLSCAASGFT      FSDTAMMWVR      QAPGKGREWV      AAIDTGGGYT

         70              80              90             100             110             120
YYADSVKGRF      TISRDNAKNT      LYLQMNCLKP      EDTARYYCAK      TYSGNYYSNY      TVANYGTTGR

        130             140
GTLVTVSSAA      AGLEHHHHHH
```

### Nanobody expression and purification

Plasmid encoding the DNA sequence for anti-mouse κLC-Nb with Ser-to-Cys mutation in position 87 was purchased from GenScript with a C-terminal 6xHis-tag in pET22b(+) expression vector carrying ampicillin (AMP) resistance. The Nb was expressed in the E.coli strain BL21 (D3) (GenScript).

In brief, a single colony was transferred to 20 mL 2xTY medium supplemented with AMP at a final concentration of 100 μg/mL in a 100 mL conical flask and grown over night (O/N) (16 h) at 37 °C, 220 RPM. The following day, the O/N culture was transferred to a 5 L Erlenmeyer flask containing 2 L terrific broth (TB) medium supplemented with 200 mL salt buffer (0.17 M $KH_2PO_4$, 0.72 M $K_2HPO_4$, pH 7), 2 mL $MgCl_2$ (2 M), 20 mL 10% (w/v) glucose solution and 2 mL AMP (100 mg/mL). The culture was grown at 37 °C and 120 RPM to an optical density 600 (OD600) ~0.6 at which point the expression was induced with 0.5 mM isopropyl β-D-1-thiogalactopyranoside (IPTG) at 25 °C O/N. The following day, cells were harvested and resuspended in lysis buffer (PBS, 400 mM NaCl, 20 mM imidazole) supplemented with

553549. Lot. 5274525), was Azide-activated using the GlyCLICK Azide activation kit (Genovis L1-AZ1-025) according to the manufacturer's instructions. In brief, 300 μg mIgG2a was concentrated to 1 mg/mL TBS buffer. The antibody was de-glycosylated by head-over-head rotation for 30 min at RT in the GlyCLICK column. The de-glycosylated mIgG2a was separated from the column and incubated with Gal-Az O/N at 30 °C. Azide-activated mIgG2a was concentrated and washed with 6 mL TBS buffer in 100 kDa molecular weight cutoff (MWCO) Amicon (Merck Millipore, cat: UFC210024). The Azide-activated antibody at a concentration of 0.8 mg/mL was conjugated with 7x excess of DBCO-S1-handle with an extension carrying P3 (Table S1) docking site (Metabion Inc.) by incubating O/N at RT and protected from light. To remove unreacted IgG as well as DNA, mIgG2a-DNA-purification was performed on an ÄKTA pure system (Cytiva) using 1 mL RESOURCE Q (Cytiva, cat: 17-1177-01) column. Appropriate fractions were concentrated and buffer-exchanged into PBS via 100 kDa MWCO centrifugal filter. The mIgG2a-DNA conjugate was stored at 4 °C at a concentration of 3.3 μM.

## Immunofixation of B1-8hi B cells for DNA-PAINT

MACS-purified B1-8hi B cells were fixed with pre-heated (37 °C) 4% paraformaldehyde (PFA) for 30 min at RT. After fixation, cells were washed two times with PBS and transferred to 6-channel μ-Slide VI 0.5 glass bottom (Ibidi). Cells were immobilized on the glass surface by centrifugation at 1000 g for 15 min in a swinging bucket rotor using in-house built adaptors. Channels were rinsed with PBS to remove unbound cells and subsequently incubated with 0.1 M NH$_4$Cl$_2$ solution for 5 min followed by 2× wash steps. Cells were permeabilized with 0.1% Triton X-100 (Sigma-Aldrich) for 5 min at RT followed by 2x wash. To assure cells remained immobilized to the glass surface, the slide was centrifuged once at 1000 g for 10 min. Hereafter, cells were incubated with blocking buffer (PBS, 3% BSA, 1 mM EDTA, 0.02% Tween-20, and 0.05 mg/mL salmon sperm DNA) in the dark for 1 h at RT. The DNA-modified κLC-Nb was diluted to 30 nM in blocking buffer and added to the cells for incubation at 4 °C O/N. The following day, cells were washed 3x with PBS and centrifuged at 1000 g for 10 min prior to post-fixation with pre-heated (37 °C) 4% PFA for 5 min at RT and followed by 2x wash. Prior to imaging, gold nanoparticles (Cytodiagnostics, cat: G-90-100) diluted 1:3 in buffer C (PBS, 500 mM NaCl) were introduced and incubated for 5 min before washing with buffer C.

## DNA-origami preparation

For folding DNA origami, 10 nM M13 single-stranded DNA scaffold, 100 nM of each staple, 150 nM of staples with S1 extension, 150 nM of S1HP3H mix and 250 nM of biotin-conjugated staples were pooled (Table S2) into 50 μl of 10 mM Tris-HCl, pH 8.0, 0.2 mM EDTA, 150 mM NaCl, 10 mM MgCl$_2$ buffer. Structures were thermally annealed in a Thermocycler (Eppendorf Inc.) by gradual cooling the mixture at a rate of 1 °C per 3 min from 60 °C to 4 °C. The folded origami structures were then purified from excess staples using 50 kDa MWCO (UFC205024) centrifugal filters. Purified origami structures were stored in buffer C at −20 °C until usage.

## Preparation of PEG-passivated surfaces

Poly-ethylene-glycol (PEG) passivated microscope coverslips were prepared according to the following procedure. Briefly, the coverslips and glass slides were sonicated in acetone, water, and 1 M KOH successively for around 20 min each. Both the coverslips and glass slides were then treated with Piranha solution (3:1 ratio of H$_2$SO$_4$ and H$_2$O$_2$) for 30 min. After this, they were washed with MilliQ and sonicated in methanol for 20 min before placing them in aminosilanization solution composed of 20:1:2 of methanol:acetic acid:3-aminopropyl trimethoxysilane (Roth, cat: 2328.1). Both the coverslips and slides were washed twice with methanol, followed by MilliQ water, and dried with dry N$_2$ gas. 75 mg of NHS-ester mPEG (5,000 Da, LaysonBio, cat: MPEG-SVA-5000-1g) and 25 mg of biotinylated NHS-PEG (5000 Da, LaysonBio, cat: Biotin-MPEG-SVA-5000-1g) were dissolved in a freshly prepared. 0.1 M Sodium Bicarbonate buffer with pH 8.5. 80 μl of this solution was placed on a glass slide that was lying flat, over which a coverslip was placed, and the sandwich was incubated O/N in dark and humid conditions. On the next day, the coverslips and slides were washed with MilliQ water and dried with dry N$_2$ gas. The coverslips and slides were then reacted with Methyl-(PEG)4-NHS-ester (333 Da, Invitrogen, cat: 22341) in 0.1 M Sodium Bicarbonate buffer for around 1 h. After rinsing with MilliQ and drying with N$_2$ gas, coverslips and slides were stored at −80 °C until usage (typically less than 3 months).

## Immobilization of DNA origami for DNA-PAINT imaging

Flow chambers were assembled by sandwiching double-sided tape in between PEG-passivated glass slide and cover slip with a volume of around 10 μL. Flow chambers contained 1 mm holes at each end for buffer exchange. A tubing with syringe was inserted at one end and sealed with 5-min epoxy. A 20–200 μL pipette tip was inserted at the other end for introducing buffers. First, a 0.1 mg/mL solution of neutravidin (Invitrogen Inc.) in 3% BSA in PBS was introduced into the biotin-PEG/PEG passivated flow cell and incubated for 5 min. Excess neutravidin was washed off with PBS. 250 pM DNA-origami structures were added in 100 μl Buffer B + (5 mM Tris-HCl, 10 mM MgCl$_2$, 1 mM EDTA and 0.05% (vol/vol) Tween 20 at pH 8.0) per channel, incubated for 15 min and washed with 1 mL buffer C. Gold nanoparticles (Cytodiagnostics, cat: G-90-100) diluted 1:10 in buffer C were introduced and incubated for 5 min before washing with buffer C. The mIgG2a antibody, conjugated with the S1-handle with P3 docking site (S1-P3, Table S2) was incubated at a concentration of 60 nM in 0.5× buffer C + 1.5% BSA in PBS for 15 min. Excess mIgG2a-DNA conjugate was washed off with 2 × 200 μl 0.5x buffer C. For the negative control, instead of the mIgG2a, 60 nM of the S1-P3 handle alone was incubated for 15 min and the following steps remained the same. After washing off the primary probe, the κLC-Nb-7xR3 was added in 0.5x buffer C + 1.5% BSA and incubated for >45 min at RT. Excess κLC-Nb was removed by washing with 2 × 200 μl 0.5× buffer C, 10 min before imaging.

## Microscope setup and DNA-PAINT image acquisition

Fluorescence imaging was carried out on an inverted microscope (Nikon Instruments, Eclipse Ti2) with the Perfect Focus System, applying an objective-type TIRF configuration equipped with an oil-immersion objective (Nikon Instruments, Apo SR TIRF ×100, NA 1.49, Oil). The 561 nm laser (MPB Communications, 2 W, DPSS system) used for excitation was passed through a cleanup filter (Chroma Technology, ZET561/10) and coupled into the microscope objective using a beam splitter (Chroma Technology, ZT561rdc). Fluorescence was spectrally filtered with an emission filter (Chroma Technology, ET600/50 m and ET575lp) and imaged using a sCMOS camera (Andor, Zyla 4.2 Plus) without further magnification, resulting in an effective pixel size of 130 nm (after 2 × 2 binning).

DNA-PAINT is a localization-based SR technique, in which the dye-labeled DNA-imager strands transiently bind to DNA docking strands that are stably attached to the molecule of interest. During the time of transient DNA-DNA binding, enough photons can be harvested at the same location to fit this diffraction-limited spot as a point source, while the freely diffusing imager strands are detected as background. For BCR imaging, a field of view (FOV) with a size of 512 × 512 pixels was selected, and images were acquired using a laser power of 21 mW at the objective with 40,000 frames and an integration time of 100 ms, and 80 pM of Cy3b-labeled imager strands supplemented with PCA/PCD/Trolox (PPT) oxygen scavenging system[59].

To determine the labeling efficiency of κLC-Nb, imaging was carried out in two successive rounds: First, the κLC-Nb-7xR3 was imaged by introducing 0.5 nM R3-Cy3b in buffer C + PPT into the channel. Then, Exchange PAINT was performed by first washing with 3 mL buffer C and then adding 2.5 nM P3-Cy3b in buffer C + PPT. Both imaging rounds were acquired with 561-nm excitation for 10,000–15,000 frames at an acquisition rate of 10 Hz and a power density of 143 W/cm².

## Image analysis

Image stacks were reconstructed using Picasso Localize[59]. Spots were detected based on a net gradient of 2000 and localizations were fitted to a Gaussian distribution using the least squares method. Drift correction was performed in two steps using Picasso Render first with redundancy cross-correlation (RCC) and second by picking at least 3 gold nano particles. Resulting images were filtered for high-precision localizations by rejecting localizations with fitted PSF sigma values >1.4 times the camera pixel size. After image reconstruction and filtering we obtained a Nearest Neighbor based Analysis (NeNa) localization precision[60] of 8.8 ± 1.7 nm.

## DBSCAN clustering analysis

For BCR cluster analysis, representative parts covering the majority of the B cell surface were saved as a circular or rectangular region of interest of $4–11\,\mu m^2$. Clusters were analyzed with the DBSCAN method[25]. DBSCAN identifies localization clouds by detecting a minimal number of localizations within a circle of a defined search radius. Based on the measured localization precision, we used a radius of 14 nm and a minimum number of localizations of 15.

To calculate the labeling efficiency of κLC-Nb, the channels for IgG2a-P3 and κLC-Nb-R3 were aligned using gold nanoparticles on the surface. Then, for both of the channels, DNA-origami with at least three P3-sites present were picked and further analyzed (>5000 structures). The picked DNA origamis were subjected to cluster analysis separately. Single P3-/R3-binding sites were identified based on user-defined input: the minimum number of localizations of 10–15 defining a cluster and the cluster radius of 10 nm. The center of mass of a single-molecule was detected and the localizations within the given radius were included in the clustering group. After overlaying the channels, R3 single-molecule clusters that were in close proximity (<19 nm) to P3-clusters were detected. By dividing the number of spots that had both a κLC-Nb-R3 as well as an IgG2a-P3 in close proximity by the number of spots that had only IgG2a-P3 and no κLC-Nb-R3 present, we were able to calculate the relative labeling efficiency.

## Simulation of single BCR molecules

A DNA-origami carrying two 7xR3 extensions with 5 nm distance was generated with Picasso Design. 935 of such DNA structures were simulated in a $32 \times 32$ px field of view, using the same conditions as used in imaging. Images were reconstructed with Picasso Localize and visualized in Picasso Render. DBSCAN as well as qPAINT analysis was performed according to the methods described for the BCRs.

## Quantitative PAINT analysis

To determine the number of molecules within one cluster, we made use of the programmable kinetics of DNA binding and unbinding and performed qPAINT. Influx rates $\xi$ were calibrated from at least 60 single binding sites ($n = 1$), picked per FOV. Dark times $\tau_D$ were determined by fitting a cumulative distribution function (CDF) individually for each SBS. The mean dark time $\tau_D^*$ was calculated for each FOV and used for influx rate calibration according to Eq. 1.

$$n = \frac{1}{\xi \cdot \tau_D^*} \qquad (1)$$

From the measured $\tau_D^*$, we could calculate the SBS-on-rate according to Eq. 2. We obtained good agreement of experimental and theoretical conditions when comparing the SBS on-rate in our images ($k_{on}^{SBS} = (45 \pm 15) \cdot 10^6 \frac{1}{Ms}$) to that of theoretical R3-kinetics from DNA-origami measurements ($k_{on}^t = 36 \pm 2.7 \cdot 10^6 \frac{1}{Ms}$), demonstrating appropriate measures for SBS calibration[61].

$$\xi = k_{on} \cdot c_i \qquad (2)$$

In the same way as for the SBS-calibration, every group of localizations identified by DBSCAN cluster algorithm was assigned a dark time by CDF-fitting from which the number of molecules could be estimated[26].

In order to exclude traces with unspecific imager interactions, prior to SBS calibration as well as qPAINT analysis, we implemented mean frame as well as standard deviation filters. These are based on the theory that specific DNA-PAINT localizations are indicated by repetitive transient binding events that are evenly distributed over the acquisition time, leading to a mean frame of around half the number of total frames acquired. Unspecific binding is indicated by a disproportional accumulation of binding events in a short amount of time, giving rise to a mean frame shifted towards lower or higher frame numbers. To remove these localization clouds, we fitted the mean frame value of all detected localization clouds, and the cutoff value was set at plus or minus the standard deviation. Clustered events occurring around the range of the mean frame cannot be removed by mean frame filtering, which is why we additionally applied a standard deviation frame filter. As clustered events result in a mean frame located within the frames of their random appearance, the resulting mean frame standard deviation is small, and therefore we excluded all clusters with a mean frame standard deviation <4000 frames[62,63].

## Sequences

Q1: 5′-NH$_2$-C6-mC + CmG + TmCmCmT + GmA + GmCmC-3′
Q2: 5′-NH$_2$-C6-mCmA + CmA + GmTmG + GmA + CmGmG-3′
Q3: 5′-NH$_2$-C6-mG + GmC + TmCmAmCmC + GmA + TmC-3′
Q4: 5′-NH$_2$-C6-mGmA + TmC + GmGmAmC + TmG + TmG-3′
8-7 dumbbell: 5′-CAGACGTCAATCAGAGACGTCTGTTT<u>CACAGTCCGA</u><u>TC</u>−3′
8-7-8 dumbbell: 5′-CAGACGTCAATCAGAGACGTCTGCTCACACCCATC
AGCGGTGTGA
GTTT<u>CACAGTCCGATC</u> −3′
12-7-12 dumbbell: 5′-CGCTCATACGCGCATCAGCCGCGTATGAGCGCA
CGTATGAGCGC
ATCAGCCGCTCATACGTGTTT<u>CACAGTCCGATC</u>−3′
15-7-15 dumbbell: 5′-CGACTCATACGATCGCATCAGCCGATCGTATGAG
TCGCACGTCA
TGAGATCGCATCAGCCGATCTCATGACGTGTTT<u>CACAGTCCGATC</u> −3′

LNA and 2′-Ome RNA nucleotides are indicated with "+" and "m", respectively. DNA nucleotides are indicated in capital letters, and underlined nucleotides indicate region complementary to the HJ Q4 strand. Q1, Q2, Q3, and Q4 oligos were synthesized with a 5′ amino modification via a carbon 6 (C6) linker by the lab of Jesper Wengel, University of Southern Denmark[29]. DNA oligos were purchased from Integrated DNA Technologies (IDT).

## Conjugation reactions

Q oligos were conjugated to the following two haptens, 4-hydroxy-3-nitrophenylacetyl-O-succinimide ester (NP-Osu) and 4-hydroxy-3-iodo-5-nitrophenylacetyl amino caproyl acetyl-O-succinimide ester (NIP-eCAP-Osu) (Santa Cruz Biotechnology), Alexa Fluor 647 (AF647) or Atto488 NHS ester (Thermo Scientific). The reactions were carried out in 100 mM HEPES (pH 8.2), ~40% DMSO in a 40-fold excess of the NHS ester at 25 °C, 500 RPM shaking for 4 h.

## Reverse-Phase HPLC purification

All reaction products were ethanol precipitated, reconstituted in nuclease free water and subsequently purified by RP-HPLC (Agilent) on a Phenomenex Evo C18 reverse-phase column using a MeCN gradient starting from 5% MeCN, 5% triethylammonium acetate (TEAA) to 95% MeCN over 25 min followed by 6 min with 95% MeCN. Based on UV detection at wavelengths 260 nm and 430 nm the fractions holding the reaction products were collected and lyophilized, and reconstituted in nuclease free water.

## Electrophoretic mobility shift assays

Reaction products were analyzed using polyacrylamide gel electrophoresis (PAGE). Denaturing PAGE gels (12%) were prepared by mixing 17 mL UreaGel Diluent (National Diagnostics), 17 mL UreaGel

Concentrate (National Diagnostics), and 4 mL 10x Tris-Borate EDTA (TBE) (Gibco by Life Technologies), 320 μL ammonium persulfate (APS) (1%), and 16 μL tetramethylethylenediamine (TEMED), both purchased from Sigma-Aldrich. Gels were prerun for 30 min at 20 W before loading the samples. Native PAGE gels (12%) contained 12 mL AccuGel 29:1 (40%) (National Diagnostics), 2 mL 10× TBE, 26 mL MilliQ, 320 μL APS, and 16 μL TEMED. All PAGE gels were stained with 5 μL SYBR Gold (Invitrogen) in 200 mL MilliQ water for 15 min and scanned on an Amersham Typhoon 5 scanner (GE Healthcare). An Ultralow range DNA ladder (10–300 bp, Thermo Scientific) was used as a size marker. NuPAGE 4–12% Bis-Tris SDS-PAGE gel (ThermoFisher) was run in MES buffer (ThermoFisher) at 150 V for 40 min. Subsequently, the gel was stained in Coomassie Brilliant Blue for 1 h and destained in MilliQ water O/N before scanning on a GelDoc Ez Imager (Bio-Rad).

### Liquid chromatography-mass spectrometry (LC-MS)

LC-MS analysis was conducted to find the mass and assess the purity of the conjugates. This was done by analyzing 40 pmol of each conjugate using a Shimadzu LC-MS-2020 (ES) coupled to a Shimadzu Prominence RP-UPLC system equipped with a Phenomenex Gemini 3 μm C18 column 100 ×4.6 mm in a gradient of MeOH in HFIP (100 mM)/TEA (80 mM) buffer. HFIP, 1,1,1,3,3,3-hexafluoro-isopropanol; TEA, trimethylamine.

### Assembly of HJ and DNA dumbbell constructs

The assembly of HJs was conducted by mixing equimolar amounts of each oligo in Dulbecco's Phosphate Buffered Saline (DPBS) or in binding buffer for the kinetic studies. DNA dumbbell structures were annealed with or without NIP-modified Q4 oligo in equimolar ratio in DPBS. In a Thermal Cycler (Eppendorf), the samples were heated up to 70 °C and cooled down to 4 °C by following a linear temperature ramp over 90 min.

### Bio-layer interferometry binding assay

Binding studies of NP/NIP-functionalized HJs were carried out using an Octet platform (FortéBio). Protein A biosensors (FortéBio) were used for the immobilization of recombinant mouse monoclonal anti-NP antibodies from B1-8 clone (Abcam; Catalog No.: ab206523, Lot. GR3248401) diluted in binding buffer (PBS, 0.02% Tween-20, 0.1% BSA, pH 7.4). The biosensors were prehydrated in binding buffer for a minimum of 10 min prior to coating with the anti-NP mAbs (1 μg/mL) through the interactions of protein A with the Fc region of the mAb. To remove excess anti-NP mAb, the sensors were washed once in binding buffer. The binding of anti-NP mAb to HJ constructs bearing 0–3 NP or NIP molecules was conducted using a 2-fold serial dilution ranging from 20 nM to 0.2 nM. The association and dissociation steps lasted 350 and 500 s, respectively. Regeneration of sensors occurred for 2 × 5 s in 10 mM glycine (pH 1.7).

### 2D binding kinetic analysis

The two-dimensional fits were made on the MATLAB 2012a platform (Mathworks) using the fitting tool EVILFIT version 3 software created by Peter Schuck[64,65]. In brief, input values matched the start and end injection time and included concentrations spanning from 219 pM to 20 nM. The association phase was fitted from t = "injection start" plus 1 s to t = "injection end" minus 5 s. Dissociation phase was fitted from t = "injection end" plus 5 s to t = "injection end" plus 200–500 s.

The operator-set boundaries for the distributions were uniformly set to limit $K_d$ values in the interval from $10^{-18}$ to $10^{-6}$ M, and $k_{off}$ values in the interval from $10^{-7}$ to $10^{-2} s^{-1}$ to ensure comparable and best quality fits reflected in a high signal to rmsd ratio.

The distribution $P(k_a, K_A)$ is calculated using the discretization of the equation:

$$R_{total} = \int_{K_{A\min}}^{K_{A\max}} \int_{k_{a\min}}^{k_{a\max}} R\left(k_a, K_A, C_{analyte}, t\right) P(k_a, K_A) dk_a\, dK_A \quad (3)$$

in a logarithmic grid of ($k_a$, $i$, $K_A$, $i$) values with 15 and 18 grid points distributed on each axis respectively. This was done through a global fit to association and dissociation traces at the above-mentioned analyte concentrations. Tikhonov regularization was used as described by Zhao et al.[66] at a confidence level of $P = 0.95$ to determine the most parsimonious distribution that is consistent with the data, showing only features that are essential to fit the data.

### Flow-induced dispersion analysis (FIDA)

Flow-Induced Dispersion Analysis (FIDA) experiments were performed using a FIDA 1 instrument employing light-emitting-diode (LED) induced fluorescence detection using an excitation wavelength of 480 nm and emission wavelength >515 nm (Fida Biosystems ApS, Copenhagen, Denmark). Non-coated capillaries with inner diameter 75 μm, outer diameter 375 μm, total length 100 cm, and length to detection window 84 cm (Fida Biosystems) were applied. Indicator samples composed of 50 or 20 nM Atto488-labeled HJs carrying 0–3 NPs or NIPs in-assay buffer (Hanks Balanced Salt Solution supplemented with 0.1% BSA, 5% FCS or anti-NP mAbs (Abcam; ab206523, Lot. GR3248401)) were analyzed using the following procedure. Initially, the capillary was flushed at 3500 mbar for 120 s. Then, assay buffer was applied at 1500 mbar for 20 s, followed by the indicator sample at 50 mbar for 10 s, and finally, assay buffer at 400 mbar for 180 s. The Taylorgrams were interpreted using the FIDA software suite, version 2.04 (Fida Biosystems ApS, Copenhagen, Denmark) and a Taylorgram fraction setting of 75%, providing the apparent hydrodynamic radius of Atto488-labeled HJs (indicator).

### Equilibrium calculations for monovalent antigens

We employed a simple model considering individual Fab arms as independent binding sites and 1:1 Ag binding. Considering that totalAg = freeAg + FabAg and totalFab = freeFab + FabAg, we can substitute in the expressions of freeAg and freeFab in the equation for the binding constant $K_D$ = freeAg × freeFab/FabAg, to achieve:

$$K_D = \frac{(totalAg - FabAg) \times (totalFab - FabAg)}{FabAg} \quad (4)$$

Abbreviating totalAg to tAg and totalFab to tFab for simplicity, and solving for FabAg/tFab, we arrive at:

$$Fraction\ bound = \frac{FabAg}{tFab}$$
$$= \frac{tAg + tFab + K_D \pm \sqrt{(tAg + tFab + K_D)^2 - 4 \times tAg \times tFab}}{2 \times tFab} \quad (5)$$

Which is equivalent to the well-established equilibrium binding equation as given in[67].

We generated an R script based on this equation with the assumption that B cells on average carry approximately 25,000 BCRs, equivalent to 50,000 Fab arms, and using standard conditions under which we had $10^6$ B cells pr. ml in 200 μl in-well volume, for a total of

200,000 B cells. This is equivalent to a total Fab arm concentration of:

$$[Fab] = \frac{50{,}000 \, Fab \, per \, B \, cell \, \times 200{,}000 \, B \, cells}{200 \times 10^{-6} \, L \, \times \, 6.022 \times 10^{23} \, molecules/mol} \qquad (6)$$

$$= \frac{1 \times 10^{10}}{12 \times 10^{19}} M = 0.083 \, nM$$

However, depending on the model, B1-8i or B1-8hi, only approximately 12% or 50%, respectively, of B cells carry a receptor with specificity for hapten.

Hence, we can define a function:

$$Fraction \, bound = function(x,y,z) \left( \frac{x + y \times a + z - \sqrt{(x + y \times a + z)^2 - 4x \times y \times a}}{2 \times y \times a} \right) \qquad (7)$$

Where $x$ is the total concentration of antigen, $a$ equals the constant $8.30 \times 10^{-11}$, which is modified by $y$, representing the fraction of cells carrying receptors specific for the antigen, and finally $z$ is the $K_D$ of the antigen-receptor pair being interrogated. Affinity constants for monovalent interactions of B1-8i Fab with NIP and NP were taken from[36] and affinity constants for B1-8hi with the same antigens were derived by assuming a 40-fold increase in affinity over B1-8i, as stated in[23]. Of note, these values are overall relatively congruent with[35].

Plotting this across the 8 orders of magnitude from $10^{-11}$ to $10^{-3}$ for B1-8hi/NIP (freqB1-8 = 0.12, $K_D = 8.25 \times 10^{-9}$), B1-8hi/NP (freqB1-8 = 0.12, $K_D = 5.5 \times 10^{-8}$), B1-8i/NIP (freqB1-8 = 0.5, $K_D = 3.3 \times 10^{-7}$), B1-8i/NP (freqB1-8 = 0.5, $K_D = 2.2 \times 10^{-6}$), yields the binding curves shown in Fig. S16.

We generated a vector of 80,000 values, distributed with 10,000 random values across each of the 8 logarithmic intervals from $10^{-11}$ to $10^{-3}$. We then calculated the fraction bound for the numbers in this vector using each of the binding curves, then calculated the mean, and finally the root mean square error (RMSE), using the formula:

$$RMSE = \sqrt{\frac{\sum_{i=1}^{n}(f1(i) - f2(i))^2}{n}} \qquad (8)$$

Where $f1$ and $f2$ refer to (7) with any specified antigen/receptor pairs and corresponding values of $y$ and $z$. Example data for one randomized vector is shown in Table S3. Taken together, this revealed the expected superiority of B1-8hi over B1-8i interactions, and of NIP over NP interactions.

To evaluate the robustness of our analyses, we asked what impact a 10% or 20% deviation in the number of B1-8 cells would have on calculated binding equilibria across the randomized vector. The results are shown in Table S4. The small RMSE resulting from even 20% deviations in the number of B1-8 cells indicated that slight experimental variations would not significantly perturb the examined activities under the given conditions.

## Flow cytometry

Relevant cells were obtained directly from lymphoid tissue or by thawing out frozen batches. Cells were diluted to $5 \times 10^6$ cells/mL and 100 µL used per well ($5 \times 10^5$ cells/well). Adjusted all-stain mixture was prepared initially, containing all dyes except for HJ variants. The dilutions of antibodies were as following: CD45R (Pacific Blue Rat Anti-Mouse CD45R (Clone RA3-6B2, BD Catalog No.: 558108, Lot. 8053605)): 1/500, CD8a (PerCP/Cyanine5.5 anti-mouse CD8a (Clone 53–6.7, Biolegend Catalog No.: 100734, Lot. B33947)): 1/300, CD4 (PerCP anti-mouse CD4 (Clone RM4-5, Biolegend Catalog No.: 553052, Lot. 0142773)): 1/300, IgMa (FITC Mouse Anti-Mouse IgM (a) (Clone DS-1, BD Catalog No.: 553516, Lot. 9015619)): 1/500, IgMb (BV650 Mouse Anti-Mouse IgM (b) (Clone AF6-78, BD Catalog No.: 742346, Lot.

0135903)): 1/300, Viability dye (Fixable Viability Dye eFluor 780 (eBioscience)): 1:2000. 144 µL of all-stain mixture was mixed with 16 µL of HJ-0/1/2/3 NP, NIP or NP-PE (NP (11)-PE (Phycoerythrin) (Biosearch Technologies), and titrations were made by taking 40 µL of the previous dilution and adding to new 120 µL of all-stain mix. 100 µL of these adjusted all-stain mixtures were then added to 100 µL of cells, and these were stained for 30 min at 4 °C. Cells were washed twice in FACS buffer and analyzed within 2 h on a NovoCyte Quanteon 4025 flow cytometer (Agilent, Santa Clara, CA) equipped with 4 lasers (405 nm, 488 nm, 561 nm and 640 nm, and 25 fluorescence detectors). Data was acquired using NovoExpress software (v. 1.5.0, Agilent) and subsequent analysis was performed in FCS Express v. 7 (De Novo Software, Pasadena, CA).

## Confocal microscopy

Frozen B1-8hi B cells were thawed out for imaging and diluted to $1 \times 10^6$ cells/mL in thawing buffer (80% DPBS, 20% FBS). 200 µL of cells were mixed with 200 µL stain buffer (PBS, 10% FCS, 0.1% Sodium Azide) containing B220 (FITC Rat Anti-Mouse CD45R (Clone RA3-6B2, BD Catalog No.: 553088, Lot. 6349700)) 1/500 dilution and HJs in described concentrations, for a final concentration of $5 \times 10^5$ cells/mL. Cells were stained for 20–30 min, washed twice with stain buffer and 200 µL of cells were added to cytospin (Shandon Elliot). Cells were spun for 3 min at 800 RPM onto SuperFrost Plus Microscopy slides and washed twice with stain buffer. After washing for 5 min, 100 µL of acetone was added to the slides in order to fix the cells. Cells were washed twice with stain buffer after fixation and mounted using SlowFade (SlowFade Diamond Antifade Mountant (Molecular Probes, Cat. No. S36967)). Imaging was performed using an Inverted Zeiss LSM710 laser scanning confocal microscope, using a 40x Plan-Neofluor (0.75 NA) objective. Images were analyzed with ImageJ.

## Calcium flux using flow cytometry

Cells for calcium flux were either freshly purified from spleens or thawed out from frozen batches. Frozen cells were thawed out in B cell medium (BCM) (RPMI-1640 supplemented with 10% FCS, 55 µM 2-Mercaptoethanol, 1% P/S (100 Units/mL Penicillin, 100 µg/mL Streptomycin), 10 mM HEPES, 1 mM Sodium Pyruvate, and 1% MEM NEAA) warmed at 37 °C. Cells were diluted to $1 \times 10^6$ cells/mL in Hank's Balanced Salt Solution (HBSS) (Gibco, Cat. No. 14025050). A total of $3 \times 10^6$ cells in 3 mL were incubated with 3.5 µM of Fluo-3 (AM) (Invitrogen, Cat. No. F1241) and 7.3 µM of Fura Red (AM) (Invitrogen, Cat. No. F3021) for 30–45 min at 37 °C[68]. After the incubation, the cells were washed once with HBSS and resuspended in 3 mL to a final concentration of $1 \times 10^6$ cells/mL. Of these, 240 µL of cells were used per flux measurement and 30 µL were used to measure the background signal, before addition of stimulant. Goat Anti-Mouse IgG (H + L), unconjugated, highly cross-adsorbed and biotinylated (Invitrogen, Cat. No. A16080, Lot. 50-66-042816) was used as a positive control for B1-8hi B cells, whereas purified Rat Anti-Mouse Ig λ1, λ2 & λ3 Light Chain (Clone R26-46 (RUO), BD Bioscience Catalog No.: 553432, Lot. 8243749) was used as a positive control for B1-8i B cells. HJs bearing 0, 1, 2 or 3 NP/NIP or DNA dumbbells with or without 1xNIP were studied as activators in concentrations described in the figure legends. Either 170 or 130 µL of cells were used to measure calcium flux after the addition of stimulant, at a flow rate of 35 µL/min. Cells were analyzed on a NovoCyte Quanteon 4025 (Agilent). Ratiometric analysis of Fluo-3 and Fura Red dyes was performed using FCS Express in order to determine changes in intracellular calcium levels.

## Imaging flow cytometry

Freshly MACS-purified B cells were diluted to $4 \times 10^6$ cells/mL and 500 µL were used per sample. Each sample was firstly incubated with various configurations of HJ constructs containing 0–3 NP/NIP in MACS buffer, for 0, 10 or 30 min at 37 °C. Afterwards, the cells were

placed on ice, then stained with anti-B220 (FITC Rat Anti-Mouse CD45R (Clone RA3-6B2, BD Catalog No.: 558108, Lot. 8053605)) and viability dye (Fixable Viability Dye eFluor 780 (eBioscience)) for 20 min at 4 °C and Hoechst (Nordic Biosite, Cat. No. CDX-B0030-M025) for 10 min at RT. After staining, the cells were washed once and analyzed using an ImageStream[X] MkII (Amnis, Luminex Corporation) at x60 magnification. At least 5000 NP positive B cells were recorded for each sample. IDEAS software (v.6.2, Amnis, Luminex Corporation) was used for analysis.

## Reporting summary

Further information on research design is available in the Nature Portfolio Reporting Summary linked to this article.

## Data availability

The datasets generated during the current study are available from the corresponding author on reasonable request. Raw data via data deposit on Dryad at https://doi.org/10.5061/dryad.bg79cnpfb. Source data are provided with this paper.

## Code availability

Custom code associated with the article is publicly available on GitHub (https://github.com/jungmannlab/picasso). The R script used for binding curve modeling (Fig. S16 and associated tables) is available on Zenodo via the DataDryad DOI.

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

## Acknowledgements

We are indebted to Simon Davis for insightful discussions and the first suggestion that the kinetic segregation model could be applicable to B cell activation. Anja Hauser and Thomas Winkler kindly provided the B1-8i Jk knock-out and B1-8hi models, respectively. Dirk Görlich kindly provided the anti-mouse-Kappa nanobody. We are grateful to Lisbeth Jensen and Thomas Wittenborn for expert technical assistance and Anne Færch Nielsen for critical review of our paper. We thank Charlotte Christie Petersen and Anja Bille Bohn for help with flow and imaging cytometry, and Christian Garm for assistance with imaging. This study was mainly funded by the Danish National Research Foundation through the Center for Cellular Signal Patterns (CellPAT) (DNRF135, J.K.). S.E.D. was additionally funded by a Carlsberg Foundation Distinguished Fellowship (CF18-0446, S.E.D.).

## Author contributions

S.E.D. and J.K. conceived and initiated the project. A.F., M.O., I.B., A.E., J.S.N., R.J., J.K. and S.E.D. planned the experiments. T.V.J. and S.T. provided valuable feedback on the experimental approaches. A.F., M.O., I.B., A.S.E., J.S.N., and D.M.D. carried out the experiments. A.F., M.O., I.B., A.E., J.S.N., R.J., P.S., K.J.M., T.V.J. and S.E.D. analyzed data. A.F., M.O., I.B. and K.J.M. prepared figures. A.F., M.O. and S.E.D. wrote the first draft of the paper. All authors provided critical feedback on the paper and approved it in its final form.

## Competing interests

The authors declare no competing interests.
