## [Peer Review File · Nature Communications]

Antigen footprint governs activation of the B cell receptorREVIEWER COMMENTS

Reviewer #1 (Remarks to the Author):

In this manuscript, Ferapontov et al. wanted to address two questions: 1) What is the resting state distribution of BCRs on the B cell surface, and 2) What are the minimal requirements for antigen-driven activation of the BCR? To answer these questions, the authors used 1) DNA-PAINT super-resolution microscopy to analyze the murine B cells' BCR distributions and 2) mono-dispersed Holliday junction (HJ) nano-scaffolds to investigate and minimum activation requirements. Taking advantage of the stochastic and transient nature of the kinetic binding of DNA fluorescent probes, the authors obtained cluster size distribution of the BCR on naïve resting B cells, mainly of the sizes of monomers, dimers, and low-order oligomers. This is in contrast to previous studies using other super-resolution microscopic methods, which reported larger sized clusters or even protein islands. Examining the BCR triggering conditions by precisely controlling hapten valency of the antigen conjugated HJs and systematically varied the antigen:BCR ratios, the authors found that polyvalent macro-size antigens activate B-cell efficiently, monovalent can trigger but require high stoichiometry; on the other hand, micro-size antigens cannot induce Ca²⁺ release. Overall, this manuscript is interesting and potentially paves the way towards the fundamental chemical-physical prerequisites about antigen-driven immuno-activation mechanisms. However, the authors should consider the following.

Major:

1. All triggering results were obtained using soluble antigen, usually at supersaturated concentrations, which differ from physiological activation conditions where antigens likely immobilized at a much larger surface at a low concentration engaging a handful of BCRs can activate B cells. This major difference casts doubt on whether the conclusions obtained in this study applicable to experiments done under more physiological activation conditions. The authors should address this issue in much more thorough way than it is written in the current manuscript. Better yet, it would be helpful if the authors could design an experiment the result of which would provide some kind of indication that the authors conclusions apply to more physiological activation conditions.
2. The authors identified a critical stoichiometry of 20:1 in their calcium experiment (Figure 4DC) but this concentration was not used in Figures 3 and 5 where binding/uptaking/internalization of the HJ constructs were studied. Why? Data at this concentration would provide additional functional readout for activation.
3. Is it possible that the non-specific absorption of RNA strands interplayed with the binding affinity and avidity of hapten with BCR? Can additional controls be used to decipher the antigen footprints about B cell activation. For example, the base levels of binding of Q4 oligo vs. HJs with and without NP/NIP at relatively high stoichiometry are needed to rule out the contribution of size of RNA strands in integrating hapten towards B-cell surfaces, then inducing activation.
4. The authors stated repeated that antigen binding to BCR in and of itself insufficient to drive B cell activation. This is a very interesting observation. However, the authors did not go much further; rather, they waved their hands by introducing a very ill-defined notion of antigen footprint. If this is so important such that these words appear in the title of the paper, the authors owe to the themselves and to the readers a more clearly defined concept and back it up with data.

Minor:

1. Due to the large discrepancy between the estimation of BCR per cell using DNA-PAINT and literature reported values, the authors should discuss the past super-resolution studies and direct flow cytometry data to provide possible explanations and justifications.
2. Figure 2E-J, right panel: K_d and k_{off} have units so one has to use reference units to scale them in order to take a log. Please state the units used to normalize the K_d and k_{off} values.
3. Line 299: 14,000 BCRs vs. 94,000 BCRs, based on 20:1 vs. 2,000:1 instead of 2,000:1 vs. 20:1?
4. Line 327: Fig. 5A, top instead of bottom?

Reviewer #2 (Remarks to the Author):

The paper by Ferapontov et al is an interesting addition to the attempts to crack the mechanism of BCR activation. It re-addresses questions raised many times before but with improved tools. The experiments are thoughtfully designed and the analysis is careful. This is certainly one of the best analyses of BCR distribution on the cell surface. The careful investigation of the relationship between the valency, size and affinity of antigens and B cell activation is also valuable. Overall, the data do not yield a clear model for the activation of the BCR, but do provide a logical assessment of antigen qualities that matter for it.

However, there are a number of problems with the manuscript that limit enthusiasm for the current version.

1. The DNA PAINT imaging needs to be better explained and illustrated in the figure. For example, how are the single binding sites defined? Are they single anti-Igkappa or single BCR? Can the authors provide evidence for saturation of the staining? Why were the cells permeabilised? Could this disturb the BCR organisation?
2. It is difficult to see how the DBSCAN function selects the clusters. It would be good if the figure illustrated the result of the algorithm with high magnification. This is important because the cluster area defines the BCR density, which is used as an argument that the BCRs are not physically bound to each other. On a similar note, please also clarify what is meant by cluster, island and oligomer. The section starts talking about oligomers, but it is later determined that these structures are loose groups of molecules, not actually oligomers of the BCR.
3. The use of B cells from B1-8i mice as controls is unclear. Are these mice always also Igkappa -/-? If yes it should be stated so.
5. The localisation analysis and discussion should cite the recent paper by Castro et al Nat Communication 2019, which concluded similarly using STED.
6. Binding of the HJs in vitro is done with anti-NP antibodies. As far as I can tell these are not the B1-8 clone. This makes this experiment less relevant to the studies with the B1-8 B cells. If this is used just to generally show the avidity effects then it should be made clear in the text.
7. The results of the EVILFIT analysis are difficult to understand. It may be best to derive one apparent K_d and off rate to make it clearer as the significance of the binding heterogeneity is unclear. Previous measurements of the B1-8 to NP binding were done in the Liu et al paper as cited, but these measurements likely involved multivalent interactions. Thus the offrates are quite slow and may not relate well to the B1-8 binding on the cells. It would be best to start with purely monovalent k_{on} and k_{off} , eg. as measured in Natkanski et al Science 2013.
8. How does the measured radius of the HJs measured by FIDA compare to the predicted size from the length of the DNA oligos? For considerations of the binding (e.g. whether they can engage two Fabs of the same BCR) it would be important to know the distance between the haptens. Binding to the anti-NP antibody tether to the chip in a low density could estimate the avidity effect when binding to a single BCR.
9. Since the binding to the anti-NP antibody is likely different from binding to the BCR, it may help to derive apparent K_d of the different antigens in binding to the B cells from the flow cytometry staining.
10. In the calcium signalling experiments, can the figure also show the binding of the antigens? At the moment is not clear from the figures to what extent the monovalent antigens bind under conditions when calcium is not detected (although estimates are provided). It almost seems that the signalling does scale with binding proportionally. For example, the monovalent NIP needs just 10x more than bivalent NIP to induce calcium, which is similar to the difference in avidity.
11. The antigen internalisation experiments do not add much information on the downstream events triggered by the antigens. All antigens will get internalised with the BCR, even those that

do not stimulate the BCR as there is constitutive internalisation of the receptor for recycling. In the mouse B cells, this happens very fast and involves substantial fractions of the BCRs. Intracellular phosphorylation, a real antigen processing/presentation assay, or CD86 up-regulation could reveal more.

12. The conclusion about the importance of antigen size or mass is interesting, but seems preliminary from just two types of antigens (not counting the chemical haptens themselves). Using the DNA, it should be possible to perform a "titration" of the antigen size to find out if there is a threshold or a linear relationship.

13. I am not clear from the discussion what the authors propose as the mechanisms by which the footprint of the antigen is sensed? It seems that they do not favour kinetic segregation, but also do not exclude it. Similarly, BCR aggregation is not favoured as a trigger, although it is deemed important as a mechanism to make the response more robust. On the other hand, conformational changes on an individual BCR are not discussed although they may fit the model? Can the discussion be more specific?

Reviewer #3 (Remarks to the Author):

This study employed quantitative superresolution microscopy (qPAINT) to characterise the clustering of BCRs on resting B cells, and RNA-scaffold-based antigens with well-controlled affinity and valency to determine their binding to B cells and their capacity to induce B cell activation, as measured by Ca²⁺ mobilisation. The authors conclude from their clustering analysis that a significant portion of BCRs exists as loosely-packed oligomers. As expected from the literature, it is found that higher valency antigens and/or higher affinity antigens activated B cells more strongly, with no differences in antigen internalisation. Surprisingly, monovalent antigens with the same nominal affinity but coupled either to RNA-scaffolds of different sizes or only carrying small chemical groups evoked differential responses.

The main strengths of this study are that the chosen approaches have the capacity to overcome technical limitations of previous study, both in terms of SRM counting methods, as well as in the design of the antigens with defined valency/affinity. Some of the findings, i.e. that monovalent antigens can activate BCRs and that clustering of BCRs is not required for activation, are new and make a significant contribution to the field.

However, the main two conclusions of the paper are insufficiently supported by the data. Unless further validations are performed, larger parts of the discussion remain speculative.

All experiments have been performed to highest standards, and details provided in the methods section in most cases allow for an independent reproduction.

Major points:

(1) Analysis of BCR oligomerisation state by qPAINT.

a. qPAINT calibration, picking of single binding sites (SBS) (lines 119f): How were the SBS picked? Is it assumed that a SBS contains one nanobody/single docking strand, as the name suggests, or two nanobodies/docking strands, as would be the case for an Ig molecule labelled at both light chains? If the prior is the case, was it taken into account that a BCR ideally contains 2 BS when calculating the oligomerisation state of BCRs? Or how was the potentially imperfect labelling of less than 100% determined / accounted for (according to lines 138f, it seems it wasn't)?

b. Analysis of cluster density: was this analysis limited to imaging data from the top and bottom membranes (which are approximately lying in the focal plane)? Imaging clusters at the sides of cells where the membrane is more vertical would lead to an underestimation of the cluster area. Has this been taken into account?

c. As a control, assuming a random distribution of BCRs on a spherical B cell of 7.1 μm diameter and using the expected density of 10,000 BCRs on a surface of $4\pi r^2 = 4 \times 3.14 \times 3.55 \times 3.55 \mu\text{m}^2 = 158 \mu\text{m}^2$ (see Figure 1F), simulations could be performed and analysed by the same DBSCAN algorithm to test which fraction of BCRs would be detected as clusters (false positives). This estimate could help to corroborate the authors' conclusion that BCRs apparently form loosely packed clusters. Related to this control is the question if the DB Scan analysis, which was performed on 2D coordinates, took into account that the imaging depth covers both, upper and lower membrane (see Figure 1B), e.g. by pre-filtering localisations by their z-position to restrict the analysis to one membrane only and avoid artifacts due to projection?

(2) Steric mechanism of BCR activation, antigen 'footprint' (incl. title):

The different monovalent antigens employed differ not only in their size, but also in other properties, i.e. charge, which could alter the effective affinity of the antigen construct for BCR. A comparative measurement of binding affinity similar as for HJ-1xNIP (Figure 2F) also for the other employed constructs (NIP-OH, NIP-e-aminocaproic acid, Q4-oligo) could help to clarify if a changed affinity alone is sufficient to explain the observed difference in signalling (Figure 6), without the need for steric effects.

(3) Line 246, 441ff: literature-derived estimate of $\sim 100,000$ BCRs per naïve mature B cell: source is missing. Discrepancy to the number determined by qPAINT (Figure 1F, 10,000) is discussed rather in a hand-wavy fashion. If this difference was due to incomplete labelling, the labelling efficiency would only be 10%, which would devalue the clustering analysis. If the difference was due to species differences, does quantitative proteomics data exist in the published literature which could be compared to support this hypothetical explanation?

Minor points:

Abbreviations that have not been defined (might be needed depending on the journal's policy):

Line 45: ITAM; Line 46: Src, Syk; Line 47: RAS, ERK, JNK, IP3, NFAT

The derivation/origin of 'literature' data for Figure 1F, 5.5 μm diameter is completely unclear.

Line 205: 'but not for control HJs without hapten (Fig. 3A)'. This control is not shown in Figure 3A.

Figure 3, C and D, and figure legend 3: it would be helpful to state in the figure legend that part C refers to NP and part D to NIP since the colors of are not explained and might confuse the reader, nor are the two graphs discussed separately in the results text.

REVIEWER COMMENTS

Reviewer #1 (Remarks to the Author):

In this manuscript, Ferapontov et al. wanted to address two questions: 1) What is the resting state distribution of BCRs on the B cell surface, and 2) What are the minimal requirements for

antigen-driven activation of the BCR? To answer these questions, the authors used 1) DNA-PAINT super-resolution microscopy to analyze the murine B cells' BCR distributions and 2) mono-dispersed Holliday junction (HJ) nano-scaffolds to investigate and minimum activation requirements. Taking advantage of the stochastic and transient nature of the kinetic binding of DNA fluorescent probes, the authors obtained cluster size distribution of the BCR on naïve resting B cells, mainly of the sizes of monomers, dimers, and low-order oligomers. This is in contrast to previous studies using other super-resolution microscopic methods, which reported larger sized clusters or even protein islands. Examining the BCR triggering conditions by precisely controlling hapten valency of the antigen conjugated HJs and systematically varied the antigen:BCR ratios, the authors found that polyvalent macro-size antigens activate B-cell efficiently, monovalent can trigger but require high stoichiometry; on the other hand, micro-size antigens cannot induce Ca²⁺ release. Overall, this manuscript is interesting and potentially paves the way towards the fundamental chemical-physical prerequisites about antigen-driven immuno-activation mechanisms. However, the authors should consider the following.

Authors: We thank the Reviewer for the overall positive reception and the recognition that our work may pave the way towards an understanding of the fundamental chemical-physical prerequisites for antigen-driven activation.

Major:

1. All triggering results were obtained using soluble antigen, usually at supersaturated concentrations, which differ from physiological activation conditions where antigens likely immobilized at a much larger surface at a low concentration engaging a handful of BCRs can activate B cells. This major difference casts doubt on whether the conclusions obtained in this study applicable to experiments done under more physiological activation conditions. The authors should address this issue in much more thorough way than it is written in the current manuscript. Better yet, it would be helpful if the authors could design an experiment the result of which would provide some kind of indication that the authors conclusions apply to more physiological activation conditions.

Response: We acknowledge the Reviewer's very valid point, that activation conditions in vitro may differ from those in vivo. As we also explicitly state in the manuscript:

"The activity of larger monovalent antigens is highly concentration-dependent, and robust activation requires what is arguably supraphysiological levels, although on a side note, it is not really known what the antigen availability is in vivo, and what range of relative ratios of antigen to cognate BCRs is encountered in the physiological setting. In our in vitro setting, monovalent antigen activity required molar stoichiometries that indicated engagement of a significant number of BCRs at the conditions employed."

That being said, we have now re-calculated the occupancy levels based on the K_D values reported in Natkanski et al. (Natkanski et al., 2013), as suggested by Reviewer 2's point 7. Furthermore, we have taken into consideration our more accurate assessment of BCR levels on the B cells (based on our measured labeling density and a correction factor for BCR surface area taking into considerations membrane ruffles – see details in response to separate point). Taken together, this indicates that our occupancy is in fact significantly lower. Thus, we are less on the supraphysiological side in terms of actual receptor occupancy than initially estimated.

While it is true that our in vitro scenario may not precisely reflect the in vivo scenario, what allows us to begin to interrogate the fundamental chemical-physical prerequisites for antigen-driven activation in the first place is exactly the reductionist approach: if we want to know the mechanism behind how, at a molecular level, antigen can elicit BCR activation, we have to isolate this molecular event. By defining the minimal requirements for the antigen, we believe that we are achieving exactly this isolation of the molecular event. The most parsimonious interpretation of our observations is that the mechanism we define would be the same mechanism that is operational in recognition of more complex antigens, or antigens bound on surfaces of cells in vivo. We consider this particularly likely, because the proposed mechanism can easily explain how antigen recognition in these cases would be more efficient, and furthermore we can rationalize why this would make sense evolutionarily: i.e., “bigger is bad’er” – a hapten or any other small molecule is likely not a threat in an immunological sense, but larger entities, such as a bacterial toxin, prions, viruses or bacteria, likely are. This is in fact the argument we arrive at in the discussion:

“Soluble antigen recognition is arguably the simplest scenario of B cell activation, and that which sets it apart from T cell recognition of peptide MHC; yet in vivo, the BCR activation mechanism must at the same time accommodate antigen scaffolding on membranes, such as the surface of follicular dendritic cells^{20,56,57}. The notion that size and valency are important determinants of antigenic activity is additionally well in line with the evolutionary drive to recognize larger particulate material, such as virus particles or bacteria, and to be ignorant of smaller, likely innocuous antigens. It is also congruent with the long-standing notion that haptens do not elicit immune responses in and of themselves. The higher potential of repeated ligands can be rationalized in terms of the common properties of biologically relevant antigens: sizeable particles with repeated structures, such as the repeated carbohydrate structures of the glycocalyx or spike proteins of the virion. Yet the capability to respond to sizeable monovalent antigen averts vulnerabilities in the B cell defense.”

Unfortunately, we are not aware of any experimental design with currently available technologies that would allow us to directly verify our mechanism of activation in vivo. However, as detailed below, we have further corroborated our finding that antigen binding in and of itself does not drive activation (Figure S14), and further investigated the antigen requirements to substantiate that indeed it is the antigen footprint that governs BCR activation by soluble, monovalent antigen. These new data have been included as new Figure 7, as outlined in our response to point 4 below.

2. The authors identified a critical stoichiometry of 20:1 in their calcium experiment (Figure 4DC) but this concentration was not used in Figures 3 and 5 where binding/uptaking/internalization of the HJ constructs were studied. Why? Data at this concentration would provide additional functional readout for activation.

Response: In Figure 4, we present two stoichiometries, 7500:1 and 75:1 (note: updated ratios based on new BCR estimate and corrected rounding), termed high and low, respectively. However, we have carried out finer titration series, demonstrating a dose-dependent activity of all antigens. The data presented in Figure 4 are merely representative stoichiometry points, at which we were able to detect activation with all antigens, or selective activation with polyvalent antigens, respectively. Based on the Reviewer’s suggestion, we have now included an additional Supplementary Figure 11 showing calcium flux titration curves for HJ-3NIP (panel A), HJ-1NIP (panel B) and HJ-1NP (Panel C) antigens on B1-8hi cells as representative

examples of full titration curve data. Taken together, these data 1) definitively demonstrate that monovalent antigen can indeed activate the BCR, and 2) explain how it may be that previous studies have reported contradictory results regarding this.

Regarding the question of why we are not using the exact same stoichiometry for the experiments presented in Figures 3 and 5, the short answer is that we had not defined this 'critical stoichiometry' when carrying out those experiments. But the long answer is that there is in fact no such thing as a universal 'critical stoichiometry', it depends on the experimental conditions:

- For the calcium flux experiments, the antigen is added to a well containing the calcium dye-loaded B cells, and this is then immediately sampled on the flow cytometer. Although the B cells and the medium they are in is drawn into a sheath, there is likely minimal admixture of the medium and the sheath fluid in the time frame from entry into the SIP until the interrogation point. This means that the antigen, after addition, is present at a constant concentration, and, as seen in the binding curve in Figure 4 panel A, left, over time as we are interrogating the cells, we are approaching equilibrium binding. Activation actually occurs prior to attainment of the equilibrium, as seen in Figure 4 panel A, right. Hence, this is ideal to understand the dynamic and immediate activation signal elicited by antigen contact.
- For both the flow binding experiments and the confocal imaging experiments presented in Figure 3, we have incubated cells with the antigens on ice for a significant incubation time to allow binding, then washed multiple times, then analyzed. Hence, in this setup, antigen cannot be internalized, and we are shifting the equilibrium extensively due to the washing, which removes unbound antigen. For the monovalent, and particularly for the low-affinity NP interactions, we observe a dramatic reduction in the fraction of cells that carry antigen. Hence, this is ideal to interrogate binding to the BCRs on the surface and delineate differences in functional affinity
- For the imaging cytometry experiments presented in Figure 5, we have incubated cells with antigen either cold (0 min at 37 °C), or for extended periods (10 or 30 min) at 37 °C. This allows binding, downstream signaling and extensive cytoskeletal rearrangements that drive BCR-mediated endocytosis of the antigen. The imaging cytometer is less sensitive than the regular flow cytometer, however, this is offset by the accumulation and uptake of antigen occurring with the extended incubation times at physiological temperature. Hence, it is ideal to read out the immediate downstream functional activity of antigen binding.

For these reasons, we believe the antigen stoichiometries presented in Figures 3, 4 and 5 to be appropriate to the experimental setups and questions addressed. The full titration ranges for calcium flux experiments in new Figure S11 cover the stoichiometries presented in Figures 3 and 5, and beyond the shown titrations for HJ-1xNIP, HJ-1xNP and HJ-3xNIP are representative also of similar titrations for HJ-2xNIP, HJ-2xNP and HJ-3xNP.

3. Is it possible that the non-specific absorption of RNA strands interplayed with the binding affinity and avidity of hapten with BCR? Can additional controls be used to decipher the antigen footprints about B cell activation. For example, the base levels of binding of Q4 oligo vs. HJs with and without NP/NIP at relatively high stoichiometry are needed to rule out the

contribution of size of RNA strands in integrating hapten towards B-cell surfaces, then inducing activation.

Response: Generally, cell surfaces are overall negatively charged, so if anything, we would expect a baseline repulsion between the nucleic acid-based nanoscaffolds and the cell surface. However, we agree that it is essential to provide these important controls. We did in fact include 'naked' HJ or dumbbell structures, i.e., without NP-/NIP-antigen, in every experiment we have run. This has now been clarified in the legends for main Figures 4, 6 and 7 (previously termed background, now specified as HJ-ONP/NIP etc.). We have additionally included flow binding data for HJ-ONP/NIP at 2:1 ratio (for Figure 3), and summary calcium flux data at high ratio for B1-8hi (6000:1) and B1-8i (1200:1) cells in new Supplementary Figure 10. Additionally, calcium flux on B1-8hi cells with 7500-fold molar excess of HJ-ONP/NIP in Figure 4 did not cause calcium flux. In summary, we have not observed any non-specific binding or 'stickiness', nor associated activation, for HJ or dumbbell structures without antigen.

4. The authors stated repeated that antigen binding to BCR in and of itself insufficient to drive B cell activation. This is a very interesting observation. However, the authors did not go much further; rather, they waved their hands by introducing a very ill-defined notion of antigen footprint. If this is so important such that these words appear in the title of the paper, the authors owe to the themselves and to the readers a more clearly defined concept and back it up with data.

Response: We fully acknowledge the Reviewer's comment – this question is paramount to the central point of the paper. We have strengthened our conclusion that antigen binding in and of itself is insufficient to drive activation by showing antigen binding data in Figure 4, which corroborate that antigen can bind, without eliciting activation. We have further strengthened this with additional experiments employing small-molecule antigen NP-FITC at up to 800,000 fold molar excess, where we see clear binding but no activation (new Figure S14). To provide more clarity to the question of antigen footprint, we have synthesized an alternative series of nanoscaffolds, in which we can very precisely vary the overall size, on top of the same 'mold', allowing us to isolate exactly the footprint factor in driving activation. Basically, we have taken the NIP-conjugated Q4-oligo, which we showed in Figure 6F did not elicit B-cell activation, then added DNA dumbbells of varying size and flexibility, to vary the antigen footprint. What we find, as represented in the new Figure 7, is that there is a 'sweet spot' of activation, dictated by 'bulk' close to the binding site. Antigen on either side of this sweet spot is either too small, or too flexible, to optimally elicit activation. Hence, we find that small-molecule hapten and hapten-Q4-oligo cannot activate, whereas hapten-HJ and hapten-Q4-oligo-dumbbells can. However, the activity varies with the dumbbell configuration, such that the inflexible and relatively bulky dumbbell 8-7-8 is optimal. Interestingly, the smaller 8-7 dumbbell is nearly as active as the larger 12-7-12 dumbbell, which is more flexible owing to the extended arms. The even larger and more flexible 15-7-15 dumbbell is more active than these, but less active than 8-7-8, suggesting that the relationship between size, flexibility and BCR activation activity is complex. Importantly, these findings also demonstrate that the footprint requirement is not a peculiarity of the HJ system, but can be extrapolated to other antigenic structures.

Minor:

1. Due to the large discrepancy between the estimation of BCR per cell using DNA-PAINT and literature reported values, the authors should discuss the past super-resolution studies and direct flow cytometry data to provide possible explanations and justifications.

Response: qPAINT (as compared to dSTORM counting analyses) is not prone to overcounting artifacts based on photophysical effects of the employed dyes (Jungmann et al., 2016), thus we argue that our reported values are closer to the ground truth. To address the effect of the unknown labeling efficiency on quantification of BCR numbers, we have performed additional in vitro experiments using a unique DNA origami-based assay to quantify the labeling efficiency of the employed anti-kappaLC-nanobody in an unbiased manner (Supplementary Fig. 3). From this assay, we obtain an overall labeling efficiency of 80% for single BCR proteins. We have thus adjusted our measured number of BCRs to approx. 25,000, taking the now quantified labeling efficiency and a 2-fold correction factor for membrane ruffles (according to (Mattila et al., 2013), and also employed by others, e.g., (Maity et al., 2015)) into account. This is somewhat below the frequently stated estimate of ~100,000 receptors per cell in the literature (Klasener et al., 2014). However, experimentally determined values for BCR number on naïve, murine B cells range widely, from 12,000 based on dSTORM ((Maity et al., 2015), total number calculated by us based on reported BCR density of 80 BCR/ μm^2 given in the supplement, with same diameter and correction for membrane ruffles as for our own number) to 335,000 ((Mattila et al., 2013), sum of IgD and IgM, based on flow cytometric quantification). We have included a brief discussion of this in the main text.

2. Figure 2E-J, right panel: K_D and k_{off} have units so one has to use reference units to scale them in order to take a log. Please state the units used to normalize the K_D and k_{off} values.

Response: We apologize for the lack of clarity here. We have now specified in the revised text that the EVILFIT plots are made with K_D on the abscissa as calculated from $K_D = k_{on}/k_{off}$, and k_{off} on the ordinate, and the abundance of each type of interaction indicated on a color scale bar. The units for K_D and k_{off} were M and s^{-1} , which is now indicated in the axis legends. We believe this is the most efficient way to solve the problem pointed out by the Reviewer without adding unnecessary complexity to the legends.

3. Line 299: 14,000 BCRs vs. 94,000 BCRs, based on 20:1 vs. 2,000:1 instead of 2,000:1 vs. 20:1?

Response: We apologize and thank the Reviewer for noting this mistake. It has been corrected, although please note that we have also recalculated our occupancy levels based on the revised K_D from (Natkanski et al., 2013), so the numbers have now been updated accordingly.

4. Line 327: Fig. 5A, top instead of bottom?

Response: Again, we apologize for this and thank the Reviewer for noting this mistake. The erroneous reference to Fig. 5A 'bottom' has been corrected to 'top' in the revised manuscript.

Reviewer #2 (Remarks to the Author):

The paper by Ferapontov et al is an interesting addition to the attempts to crack the mechanism of BCR activation. It re-addresses questions raised many times before but with improved tools. The experiments are thoughtfully designed and the analysis is careful. This is certainly one of the best analyses of BCR distribution on the cell surface. The careful investigation of the relationship between the valency, size and affinity of antigens and B cell activation is also valuable. Overall, the data do not yield a clear model for the activation of the BCR, but do provide a logical assessment of antigen qualities that matter for it. However, there are a number of problems with the manuscript that limit enthusiasm for the current version.

Authors: We thank the Reviewer for the overall positive valuation of our attempt to determine the mechanism of BCR activation, and for noting that this is one of the best analyses of BCR distribution on the cell surface.

1. The DNA PAINT imaging needs to be better explained and illustrated in the figure. For example, how are the single binding sites defined? Are they single anti-Igkappa or single BCR? Can the authors provide evidence for saturation of the staining? Why were the cells permeabilised? Could this disturb the BCR organisation?

Response: We apologize for not making this clearer in the initial submission and thank the reviewer for pointing us to this potential issue. In short, we select single molecule signals from DNA-PAINT super-resolution data as explained in the manuscript. These single molecule signals represent single BCR proteins labeled by up to two anti-kappaLC-nanobodies. To address the effect of the unknown labeling efficiency on quantification of BCR numbers, we have now performed additional in vitro experiments using a unique DNA origami-based assay to quantify the labeling efficiency of the employed anti-kappaLC-nanobody in an unbiased manner (Supplementary Fig. 3). From this assay, we obtain an overall labeling efficiency of 80% for single BCR proteins. Permeabilization of the cells post-fixation is necessary to provide the best accessibility for DNA-PAINT imager strands to their complements on anti-kappaLC-nanobodies bound to BCR proteins on the cell surface. Fixation and permeabilization should not have a significant influence on the BCR organization, as we have employed this experimental approach successfully to other membrane surface receptor molecules in the past (Strauss and Jungmann, 2020; Strauss et al., 2018).

2. It is difficult to see how the DBSCAN function selects the clusters. It would be good if the figure illustrated the result of the algorithm with high magnification. This is important because the cluster area defines the BCR density, which is used as an argument that the BCRs are not physically bound to each other. On a similar note, please also clarify what is meant by cluster, island and oligomer. The section starts talking about oligomers, but it is later determined that these structures are loose groups of molecules, not actually oligomers of the BCR.

Response: We apologize for the possible confusion in how DBSCAN operates on our data. For clarification, we now include an additional figure (Supplementary Fig. 4), where we provide zoom-ins and comparisons of DNA-PAINT raw data and DBSCAN selected areas to shed more light on the DBSCAN output. The cluster area (as is common practice in the field) is defined as the convex hull of the DBSCAN detected molecules. We note, that "cluster" areas in our

definition comprise monomers, dimers, higher order oligomers (small islands) as well as (large) islands. The distinction between each group is solely made based on qPAINT molecule numbers. We note that oligomers in our notation does not imply direct biochemical interaction of BCR molecules, but rather a denser arrangement or receptors comprising a higher density compared to a complete random arrangement at a given overall molecule density. We furthermore include an additional in silico study (Supplementary Fig. 4), highlighting the fact that (given our experimental conditions) DBSCAN is unlikely to select molecules as higher order oligomers that are in fact monomers based on the ground truth simulation.

We agree that the use of the term 'oligomers' can be easily misunderstood to mean tightly associated BCRs, when in fact we meant loose groups of molecules, as the Reviewer points out. Indeed, we defined oligomers from 3 molecules onwards and all oligomers have a density that is so low that there are likely no direct inter-BCR contacts. Accordingly, in the revised manuscript, we have renamed the oligomers (3-9 molecules) to 'small islands' and everything > 10 molecules to 'large islands'.

3. The use of B cells from B1-8i mice as controls is unclear. Are these mice always also Igkappa -/- ? If yes it should be stated so.

Response: We apologize for the lack of clarity here. The line of B1-8i mice in our colony is always also Igkappa-/- and we have now clarified this in the figure legend for Figure S2.

5. The localisation analysis and discussion should cite the recent paper by Castro et al Nat Communication 2019, which concluded similarly using STED.

Response: We apologize for the oversight and have now included a citation to this study.

6. Binding of the HJs in vitro is done with anti-NP antibodies. As far as I can tell these are not the B1-8 clone. This makes this experiment less relevant to the studies with the B1-8 B cells. If this is used just to generally show the avidity effects then it should be made clear in the text.

Response: We again apologize for the lacking presentation. We used the original B1-8 clone that we purchased from abcam. The clone number is (as also written in the materials section): ab206523 <https://www.abcam.com/hapten-4-hydroxy-3-nitrophenyl-acetyl-np-antibody-b1-8-ab206523.html> (recombinant monoclonal antibody generated by cloning of the original variable regions of the B1-8 hybridoma). We have now made this clearer in the text and associated figures.

7. The results of the EVILFIT analysis are difficult to understand. It may be best to derive one apparent Kd and off rate to make it clearer as the significance of the binding heterogeneity is unclear. Previous measurements of the B1-8 to NP binding were done in the Liu et al paper as cited, but these measurements likely involved multivalent interactions. Thus the offrates are quite slow and may not relate well to the B1-8 binding on the cells. It would be best to start with purely monovalent kon and koff, eg. as measured in Natkanski et al Science 2013.

Response: We apologize for the lack of clarity in our presentation of the EVILFIT analyses. The purpose of this analysis is exactly to be able to resolve the heterogeneous binding interactions,

instead of averaging these into a single apparent K_D , because the heterogeneity is directly related to the different valency interactions. We have added a more careful description of this in the manuscript.

We thank the Reviewer for drawing our attention to the affinity constants for the purely monovalent interactions reported in Natkanski et al. (Natkanski et al., 2013). In that study, affinity constants were only given for B1-8i binding to NP and NIP. We have extrapolated to B1-8hi under the assumption that the 'hi' version of the receptor has a 40-fold increased antigen affinity compared to the 'i' version (as stated in (Shih et al., 2002)). We have adjusted our R script, the binding curves (now Figure S16) and the data in Supplementary Tables 3 and 4, as well as recalculated our receptor occupancy levels using these affinity constants, instead of as originally the data from Liu et al. A note about this has been added to the Methods section.

8. How does the measured radius of the HJs measured by FIDA compare to the predicted size from the length of the DNA oligos? For considerations of the binding (e.g. whether they can engage two Fabs of the same BCR) it would be important to know the distance between the haptens. Binding to the anti-NP antibody tether to the chip in a low density could estimate the avidity effect when binding to a single BCR.

Response: There is no solved structure for the HJ nanoscaffold and thus the exact distances between the arms are not known for this specific structure. However, it is known that the LNA and 2'OMe RNA backbone employed promote the HJ to adopt an A-configuration as in RNA duplex. This configuration is highly stabilized by the LNA modifications since this nucleic acid analog locks the sugar backbone in an N-type conformation. Based on this, the theoretical/predicted distances of the HJ arms are on the order of approx. 3.5 nm, which can be affected by coaxial stacking in the presence of divalent ions. When taking into account the carbon spacer length used to join the NP or NIP molecules, the distances between arms become approx. 6-8.5 nm, which is close to the lower spatial tolerance limit for dual Fab arm engagement within a single IgG molecule, as determined by Högberg and colleagues (Shaw et al., 2019). FIDA measurements were used to determine the monodispersity of the HJs under different buffer conditions. The hydrodynamic radius obtained with FIDA gives an indication of the average size of the HJs in solution under certain conditions. Because the HJ is not a globular structure and can form coaxial helical stacking, and thus can adopt multiple conformations, the predicted and measured sizes are not directly comparable.

9. Since the binding to the anti-NP antibody is likely different from binding to the BCR, it may help to derive apparent K_D of the different antigens in binding to the B cells from the flow cytometry staining.

Response: We appreciate the Reviewer's point. Indeed, in many cases, the affinities derived from cell-based assays are much higher than those obtained by biophysical systems like SPR on isolated proteins (Bondza et al., 2017). However, our goal here was not to define a single K_D for the binding interaction in either setting, but rather to resolve the heterogeneity in binding affinities represented by monovalent vs. multivalent antigen scaffolds. This was possible through the EVILFIT analyses of our BLI data, but unfortunately the same approach would not be applicable to flow cytometry derived agglomerate estimates of affinity constants.

10. In the calcium signalling experiments, can the figure also show the binding of the antigens? At the moment is not clear from the figures to what extent the monovalent antigens bind under conditions when calcium is not detected (although estimates are provided). It almost seems that the signalling does scale with binding proportionally. For example, the monovalent NIP needs just 10x more than bivalent NIP to induce calcium, which is similar to the difference in avidity.

Response: We thank the Reviewer for this suggestion. We have now reanalyzed the data for the B1-8hi setting, and included binding curves in revised Figure 4 (new panels D and E) as suggested by the Reviewer. As can be seen, at high stoichiometry, binding equilibrium is reached quite fast (note that due to the high concentration of antigen-AF647, the baseline also shifts dramatically up), whereas at the low concentration, it is more reminiscent of a binding curve. It is clear that calcium flux is not simply a trivial consequence of antigen binding, because at high stoichiometry (7500:1), HJ-1xNP binds to a level comparable to that of HJ-3xNIP, but the former is much poorer at activating, compared to the latter. The peculiar observation of comparable binding level of these two antigens at the 'constant high antigen availability' conditions of the calcium flux may be explained by a more limited loading capacity of the 3xNIP antigen with high affinity binding to individual BCRs. This is even more pronounced at the lower (but still saturating) antigen concentration, where this antigen again has the highest calcium flux activity.

11. The antigen internalisation experiments do not add much information on the downstream events triggered by the antigens. All antigens will get internalised with the BCR, even those that do not stimulate the BCR as there is constitutive internalisation of the receptor for recycling. In the mouse B cells, this happens very fast and involves substantial fractions of the BCRs. Intracellular phosphorylation, a real antigen processing/presentation assay, or CD86 up-regulation could reveal more.

Response: Although non-specific constitutive uptake of antigens may occur, antigen-specific uptake is estimated to be 10,000-fold more efficient (Lanzavecchia, 1990). We acknowledge the Reviewer's point that even antigen-specific uptake of monovalent antigens could simply be a consequence of binding followed by 'passive' uptake due to BCR recycling. However, this would likely be a lot slower than active BCR-triggered endocytosis. Figure 4 is in essence intended to directly address up-front a previous report suggesting that monovalent antigen can trigger BCR activation, but does not lead to internalization (Kim et al., 2006). In that paper, the authors did not see baseline internalization of monovalent antigen, despite activation. Figure 4 demonstrates that with our HJ antigen system, we observe the expected internalization downstream of receptor engagement even when the antigen is monovalent. The 'capping' or punctate appearance of antigen and subsequent bulk internalization indicated that there was no fundamental qualitative difference between BCR activation induced by poly- and monovalent HJ antigen. This has now been clarified in the text. We agree with the Reviewer that further analyses of downstream signaling, upregulation of co-stimulatory molecules, or antigen-derived peptide presentation would more exhaustively qualify the ability of the antigens to induce physiological activation. However, we believe this is beyond the scope of the present manuscript, which is to examine immediate BCR activation requirements to shed light on the molecular mechanism behind the very first step in activation.

12. The conclusion about the importance of antigen size or mass is interesting, but seems preliminary from just two types of antigens (not counting the chemical haptens themselves). Using the DNA, it should be possible to perform a “titration” of the antigen size to find out if there is a threshold or a linear relationship.

Response: We thank the Reviewer for this excellent suggestion. As also outlined in our response to Reviewer 1, to provide more clarity to the question of antigen footprint, we have synthesized an alternative series of nanoscaffolds, in which we can very precisely vary the overall size, on top of the same ‘mold’, allowing us to isolate exactly the footprint factor in driving activation. Basically, we have taken the NIP-conjugated Q4-oligo, which we showed in Figure 6F did not elicit B-cell activation, then added DNA dumbbells of varying size and flexibility, to vary the antigen footprint. What we find, as represented in the new Figure 7 is that there is a ‘sweet spot’ of activation, dictated by ‘bulk’ close to the binding site. Antigen on either side of this sweet spot is either too small, or too flexible, to optimally elicit activation.

Hence, we find that small-molecule hapten and hapten-Q4-oligo cannot activate, whereas hapten-HJ and hapten-Q4-oligo-dumbbells can. However, the activity varies with the dumbbell configuration, such that the inflexible and relatively bulky dumbbell 8-7-8 is optimal. Interestingly, the smaller 8-7 dumbbell is nearly ‘as active’ as the larger 12-7-12 dumbbell, which is more flexible owing to the extended arms. The even larger and more flexible 15-7-15 dumbbell is more active than these, but less active than 8-7-8, suggesting that the relationship between size, flexibility and BCR activation activity is complex.

As also mentioned in our response to Reviewer 1, this demonstrates that the footprint requirement is not a peculiarity of the HJ system, but can be extrapolated to other antigenic structures.

13. I am not clear from the discussion what the authors propose as the mechanisms by which the footprint of the antigen is sensed? It seems that they do not favour kinetic segregation, but also do not exclude it. Similarly, BCR aggregation is not favoured as a trigger, although it is deemed important as a mechanism to make the response more robust. On the other hand, conformational changes on an individual BCR are not discussed although they may fit the model? Can the discussion be more specific?

Response: We apologize for our lack of clarity here. Our observations are incongruent with the dissociation activation model and the simple clustering model. We cannot rule out a potential role of a conformational/allosteric effect, however, based on available structural knowledge of the BCR/antibodies (of varying isotypes), we find this very, very unlikely. A modification of the kinetic segregation model on the other hand could potentially accommodate our observations. We have tried to clarify this in the revised discussion.

Reviewer #3 (Remarks to the Author):

This study employed quantitative superresolution microscopy (qPAINT) to characterise the clustering of BCRs on resting B cells, and RNA-scaffold-based antigens with well-controlled affinity and valency to determine their binding to B cells and their capacity to induce B cell

activation, as measured by Ca²⁺ mobilisation. The authors conclude from their clustering analysis that a significant portion of BCRs exists as loosely-packed oligomers. As expected from the literature, it is found that higher valency antigens and/or higher affinity antigens activated B cells more strongly, with no differences in antigen internalisation. Surprisingly, monovalent antigens with the same nominal affinity but coupled either to RNA-scaffolds of different sizes or only carrying small chemical groups evoked differential responses.

The main strengths of this study are that the chosen approaches have the capacity to overcome technical limitations of previous study, both in terms of SRM counting methods, as well as in the design of the antigens with defined valency/affinity. Some of the findings, i.e. that monovalent antigens can activate BCRs and that clustering of BCRs is not required for activation, are new and make a significant contribution to the field.

Authors: We thank the Reviewer for the positive reception of the SRM data and the novel antigen designs, and the recognition that some of the findings are new and make a significant contribution to the field.

However, the main two conclusions of the paper are insufficiently supported by the data. Unless further validations are performed, larger parts of the discussion remain speculative.

All experiments have been performed to highest standards, and details provided in the methods section in most cases allow for an independent reproduction.

Major points:

(1) Analysis of BCR oligomerisation state by qPAINT.

a. qPAINT calibration, picking of single binding sites (SBS) (lines 119f): How were the SBS picked? Is it assumed that a SBS contains one nanobody/single docking strand, as the name suggests, or two nanobodies/docking strands, as would be the case for an Ig molecule labelled at both light chains? If the prior is the case, was it taken into account that a BCR ideally contains 2 BS when calculating the oligomerisation state of BCRs? Or how was the potentially imperfect labelling of less than 100% determined / accounted for (according to lines 138f, it seems it wasn't)?

Response: We thank the reviewer for pointing us to this potential confusing point. In short, we first select single molecule signals from DNA-PAINT super-resolution data for calibration. These single molecule signals represent single BCR proteins labeled by up to two anti-kappaLC-nanobodies. To address the effect of the unknown labeling efficiency on quantification of BCR numbers, we have performed additional in vitro experiments using a unique DNA origami-based assay to quantify the labeling efficiency of the employed anti-kappaLC-nanobody in an unbiased manner (Supplementary Fig. 3). From this assay, we obtain an overall labeling efficiency of 80% for single BCR proteins.

As the calibration represents single BCR molecules, the downstream quantification of DBSCAN-derived clustered areas should be quantitative.

b. Analysis of cluster density: was this analysis limited to imaging data from the top and

bottom membranes (which are approximately lying in the focal plane)? Imaging clusters at the sides of cells where the membrane is more vertical would lead to an underestimation of the cluster area. Has this been taken into account?

Response: We apologize for not making this clearer in the manuscript. 3D DNA-PAINT data of BCR molecules (as depicted in Figure 1b) shows that we are in fact only imaging BCR proteins on the plasma membrane close to the cover slip and do not detect BCR receptors on the “upper” membrane. We furthermore picked the central area of the membrane close to the cover slip for subsequent qPAINT analysis. As such, a potential source of error from BCR proteins on the upper membrane can be excluded.

c. As a control, assuming a random distribution of BCRs on a spherical B cell of 7.1 μm diameter and using the expected density of 10,000 BCRs on a surface of $4\pi r^2 = 4 \cdot 3.14 \cdot 3.55 \cdot 3.55 \mu\text{m}^2 = 158 \mu\text{m}^2$ (see Figure 1F), simulations could be performed and analysed by the same DBSCAN algorithm to test which fraction of BCRs would be detected as clusters (false positives). This estimate could help to corroborate the authors’ conclusion that BCRs apparently form loosely packed clusters. Related to this control is the question if the DB Scan analysis, which was performed on 2D coordinates, took into account that the imaging depth covers both, upper and lower membrane (see Figure 1B), e.g. by pre-filtering localisations by their z-position to restrict the analysis to one membrane only and avoid artifacts due to projection?

Response: We thank the reviewer for this suggestion. We now include an additional in silico study (Supplementary Fig. 4), highlighting the fact that (given our experimental conditions) DBSCAN is unlikely to select molecules as higher order oligomers that are in fact monomers based on the ground truth simulation. With regards to the “2D-ness” of the analysis, we would like to point to our response to the Reviewer’s question 1b): In fact, we only analyze the lower part of the membrane close to the coverslip.

(2) Steric mechanism of BCR activation, antigen ‘footprint’ (incl. title):

The different monovalent antigens employed differ not only in their size, but also in other properties, i.e. charge, which could alter the effective affinity of the antigen construct for BCR. A comparative measurement of binding affinity similar as for HJ-1xNIP (Figure 2F) also for the other employed constructs (NIP-OH, NIP-e-aminocaproic acid, Q4-oligo) could help to clarify if a changed affinity alone is sufficient to explain the observed difference in signalling (Figure 6), without the need for steric effects.

Response: we cannot rule out that charge plays a role in the activity of the constructs. However, we would expect that if anything, a net negative charge would decrease the affinity, due to electrostatic repulsion with the negatively charged cell surface. Hence this would be unlikely to explain the activity of the NP-/NIP-HJ constructs, compared to the lack of activity of NIP-OH, NIP-e-aminocaproic acid and NP-FITC (added in revised manuscript, Figure S14). Furthermore, the Q4-oligo, like the HJ construct, is based on nucleic acids, and hence also has a net negative charge. Nonetheless, the NIP-Q4 oligo did not display any activity, and here the only likely explanation is the size/bulk. However, we admit that our evidence for the role of the footprint, as also noted by Reviewers 1 and 2, was rather limited in the original version of

the manuscript. To further address this, we synthesized an alternative series of nanoscaffolds, in which we could very precisely vary the overall size, on top of the same 'mold', allowing us to isolate exactly the footprint factor in driving activation. Basically, we have taken the NIP-conjugated Q4-oligo, which we showed in Figure 6F did not elicit B-cell activation, then added DNA dumbbells of varying size and flexibility, to vary the antigen footprint. What we find, as represented in the new Figure 7 is that there is a 'sweet spot' of activation, dictated by 'bulk' close to the binding site. Antigen on either side of this sweet spot is either too small, or too flexible, to optimally elicit activation. Furthermore, we have qualified the notion that it is not simply the antigen binding in and of itself that elicits BCR activation, by directly measuring the amount of antigen bound in our calcium flux experiments presented in Figure 4, and by confirming binding of NP-FITC at large excess, without any concomitant activation (Figure S14).

(3) Line 246, 441ff: literature-derived estimate of ~100,000 BCRs per naïve mature B cell: source is missing. Discrepancy to the number determined by qPAINT (Figure 1F, 10,000) is discussed rather in a hand-wavy fashion. If this difference was due to incomplete labelling, the labelling efficiency would only be 10%, which would devalue the clustering analysis. If the difference was due to species differences, does quantitative proteomics data exist in the published literature which could be compared to support this hypothetical explanation?

Response: We apologize for the oversight and now cite a reference for the literature derived estimate of 100,000 BCRs per naïve mature B cell (Klasener et al., 2014) – this is but one example, and this number can also be found in textbooks, e.g., Molecular Biology of the Cell (4th edition), and many articles quote upwards of 100,000, over 100,000, or ~120,000. As also stated in our response to Reviewer 1's similar point, qPAINT (as compared to dSTORM counting analyses) is not prone to overcounting artifacts based on photophysical effects of the employed dyes (Jungmann et al., 2016), thus we argue that our reported values are closer to the ground truth. To address the effect of the unknown labeling efficiency on quantification of BCR numbers, we have performed additional in vitro experiments using a unique DNA origami-based assay to quantify the labeling efficiency of the employed anti-kappaLC-nanobody in an unbiased manner (Supplementary Fig. 3). From this assay, we obtain an overall labeling efficiency of 80% for single BCR proteins. We have thus adjusted our measured number of BCRs to approx. 25,000, taking the now quantified labeling efficiency and a correction factor of 2 for membrane ruffles (according to (Mattila et al., 2013), and also used by others, e.g., (Maity et al., 2015)) into account. This is somewhat below the frequently stated estimate of 100,000 receptors per cell in the literature (Klasener et al., 2014). However, experimentally determined values for BCR number on naïve, murine B cells range widely, from 12,000 based on dSTORM ((Maity et al., 2015), total number calculated by us based on reported BCR density of 80 BCR/ μm^2 given in the supplement, with same diameter and correction for membrane ruffles as for our own number) to 335,000 ((Mattila et al., 2013), sum of IgD and IgM, based on flow cytometric quantification). We have included a brief discussion of this in the main text.

Minor points:

Abbreviations that have not been defined (might be needed depending on the journal's policy): Line 45: ITAM; Line 46: Src, Syk; Line 47: RAS, ERK, JNK, IP3, NFAT

Response: our apologies, we have now specified the abbreviation for ITAM on the first usage. We believe the protein naming is standard and does not need to be explained. However, if we need to provide abbreviations for these, we will of course do so.

The derivation/origin of 'literature' data for Figure 1F, 5.5 μm diameter is completely unclear.

Response: We apologize for this omission. The literature-derived estimate is based on older data from John Cambier's group (Monroe and Cambier, 1983), in which lymphoblasts were sorted based on cell diameter and this was correlated back to cell cycle stage. This paper was referenced in-text but we have now added the reference in the figure legend as well for clarity. G0 cells were found to have a diameter of 4.5-5.5 μm , with increasing sizes for subsequent cell cycle stages: early G1 = 5.5-7.0 μm , late G1 and S phase = 7.0-10 μm and S, G2 and M phase = 10-12 μm . We have purified untouched, resting B cells from spleen, and therefore should have predominantly G0 cells, with diameters upwards of 5.5 μm , according to this reference. In practice, when we measure our freshly isolated cells using our cell counter, or experimentally measure our fixed cells during the DNA-PAINT imaging, we observe diameters of around 7.1-7.2 μm . We suspect that this difference is based in the different methodologies: Cell diameter determination by flow cytometric analysis of the pulse-width (time of flight) of the axial light extinction signal (using flow cytometry in the early 1980's) vs. direct observation of individual cells by light (cell counter) or super resolution microscopy. We have chosen to report values for both these diameter estimates, but for the subsequent calculations involving cell diameter, we have used our own experimentally determined value, rather than that determined by Cambier lab 40 years ago. This is also in agreement with more recent literature estimates of around 7 μm , e.g., (Reth, 2013).

Line 205: 'but not for control HJs without hapten (Fig. 3A)'. This control is not shown in Figure 3A.

Response: We apologize for this omission. We have added this control in new Supplementary Figure 10, panel A, and the reference to this has been updated in-text.

Figure 3, C and D, and figure legend 3: it would be helpful to state in the figure legend that part C refers to NP and part D to NIP since the colors of are not explained and might confuse the reader, nor are the two graphs discussed separately in the results text.

Response: We thank the Reviewer for noticing this. We have corrected the figure annotation in the revised version of Figure 3 to make it clearer.

Authors: We would like to end by once more thanking the Editor for handling our manuscript, and the three expert Reviewers for the time taken to critically evaluate our work. We are very grateful for the constructive critique and the useful suggestions for improvements, which we believe have significantly improved our manuscript.

References included in this rebuttal:

- Bondza, S., Foy, E., Brooks, J., Andersson, K., Robinson, J., Richalet, P., and Buijs, J. (2017). Real-time Characterization of Antibody Binding to Receptors on Living Immune Cells. *Front Immunol* *8*, 455. 10.3389/fimmu.2017.00455.
- Jungmann, R., Avendano, M.S., Dai, M., Woehrstein, J.B., Agasti, S.S., Feiger, Z., Rodal, A., and Yin, P. (2016). Quantitative super-resolution imaging with qPAINT. *Nat Methods* *13*, 439-442. 10.1038/nmeth.3804.
- Kim, Y.M., Pan, J.Y., Korb, G.A., Peperzak, V., Boes, M., and Ploegh, H.L. (2006). Monovalent ligation of the B cell receptor induces receptor activation but fails to promote antigen presentation. *Proc Natl Acad Sci U S A* *103*, 3327-3332. 10.1073/pnas.0511315103.
- Klasener, K., Maity, P.C., Hobeika, E., Yang, J., and Reth, M. (2014). B cell activation involves nanoscale receptor reorganizations and inside-out signaling by Syk. *Elife* *3*, e02069. 10.7554/eLife.02069.
- Lanzavecchia, A. (1990). Receptor-mediated antigen uptake and its effect on antigen presentation to class II-restricted T lymphocytes. *Annu Rev Immunol* *8*, 773-793. 10.1146/annurev.iy.08.040190.004013.
- Liu, W., Meckel, T., Tolar, P., Sohn, H.W., and Pierce, S.K. (2010). Antigen affinity discrimination is an intrinsic function of the B cell receptor. *J Exp Med* *207*, 1095-1111. 10.1084/jem.20092123.
- Maity, P.C., Blount, A., Jumaa, H., Ronneberger, O., Lillemeier, B.F., and Reth, M. (2015). B cell antigen receptors of the IgM and IgD classes are clustered in different protein islands that are altered during B cell activation. *Sci Signal* *8*, ra93. 10.1126/scisignal.2005887.
- Mattila, P.K., Feest, C., Depoil, D., Treanor, B., Montaner, B., Otipoby, K.L., Carter, R., Justement, L.B., Bruckbauer, A., and Batista, F.D. (2013). The actin and tetraspanin networks organize receptor nanoclusters to regulate B cell receptor-mediated signaling. *Immunity* *38*, 461-474. 10.1016/j.immuni.2012.11.019.
- Monroe, J.G., and Cambier, J.C. (1983). Sorting of B lymphoblasts based upon cell diameter provides cell populations enriched in different stages of cell cycle. *Journal of Immunological Methods* *63*, 45-56. 10.1016/0022-1759(83)90208-9.
- Natkanski, E., Lee, W.Y., Mistry, B., Casal, A., Molloy, J.E., and Tolar, P. (2013). B cells use mechanical energy to discriminate antigen affinities. *Science* *340*, 1587-1590. 10.1126/science.1237572.
- Reth, M. (2013). Matching cellular dimensions with molecular sizes. *Nat Immunol* *14*, 765-767. 10.1038/ni.2621.
- Shaw, A., Hoffecker, I.T., Smyrlaki, I., Rosa, J., Grevys, A., Bratlie, D., Sandlie, I., Michaelsen, T.E., Andersen, J.T., and Hogberg, B. (2019). Binding to nanopatterned antigens is dominated by the spatial tolerance of antibodies. *Nat Nanotechnol* *14*, 184-190. 10.1038/s41565-018-0336-3.
- Shih, T.A., Roederer, M., and Nussenzweig, M.C. (2002). Role of antigen receptor affinity in T cell-independent antibody responses in vivo. *Nat Immunol* *3*, 399-406. 10.1038/ni776.

Strauss, S., and Jungmann, R. (2020). Up to 100-fold speed-up and multiplexing in optimized DNA-PAINT. *Nat Methods* *17*, 789-791. 10.1038/s41592-020-0869-x.

Strauss, S., Nickels, P.C., Strauss, M.T., Jimenez Sabinina, V., Ellenberg, J., Carter, J.D., Gupta, S., Janjic, N., and Jungmann, R. (2018). Modified aptamers enable quantitative sub-10-nm cellular DNA-PAINT imaging. *Nat Methods* *15*, 685-688. 10.1038/s41592-018-0105-0.

REVIEWER COMMENTS

Reviewer #1 (Remarks to the Author):

The authors have responded largely my concerns and the manuscript has been improved. I only have the following remaining comments.

Regarding the response to comment 1- I still doubt that the recalculated molar ratio is anywhere close to physiological. The authors argued that no one really known what the antigen availability is in vivo. But there is good information about the antigen availability for TCR in vivo. Do the authors really think there are such high excess of antigen compared to BCR?

Regarding the response to comment 4- The authors updated Figure 4 and added Figure S14 to show that NP-FITC with various molar excess could not activate properly B cell signaling. I appreciate that the relationship between size, flexibility and BCR activation activity is complex. Can the authors show this in a quantitative way using the data in Fig 7? For example, plots of Activation (fold-change of Ca²⁺ signaling) vs antigen size (gyro radius) and flexibility (order of rotational symmetry) would help for readers to visualize 'quantitative footprint-requirement of antigen to properly activate B cell'.

Reviewer #2 (Remarks to the Author):

The new manuscript is greatly improved and generally addresses my comments. I also enjoyed the clearer discussion. I only have a few minor issues:

1. The description of the binding of the HJ-Ag structures to the antibody and cells (pages 9-15) may be condensed as it only serves as a control and does not produce any novel information.

2. Line 326 is unclear. At this point, the data still show a correlation between binding and activation, because I don't see much binding detected for the monovalent HJ at concentrations that do not activate calcium flux (Fig 4E). This point becomes clearer later with the low-MW antigens. Perhaps lack of clarity also comes from the next issue.

3. Fig 4E and Fig 7 would benefit from different colors of binding/calcium lines. It is very difficult to distinguish the shades of grey or orange.

Reviewer #3 (Remarks to the Author):

The revised manuscript by Ferapontov et al. includes new data to elucidate the concept of the antigen footprint, as well as several control experiments and changes to the manuscript text to clarify technical questions raised by all reviewers. This revision constitutes a significant improvement, however, minor issues remain which, if adequately addressed, could further support the data interpretation.

Regarding Point 1.b raised during my original review: thank you for clarifying that only the lower membrane is being imaged. The confusion arose from the xyz 3D view in Figure 1 B, lower panel, which shows a difference in height (z) of ca 700 nm, while the length of the line profile is ~6 μ m. The inner ~4 μ m show signals at the bottom (blue) but also higher up (yellow and red) – what are these latter signals? Are these BCRs on membrane ruffles? A better description of this Figure in the results section would be helpful to avoid confusion of the readers. If these are membrane ruffles, are these (fully) contained in the imaging depth of the qPAINT measurements? In that case it would technically speaking not be correct to use a correction factor of 2 for the extrapolation of the BCR count per cells.

The concept of antigen footprint still remains vague. The authors have performed additional experiments with different dumbbell shaped antigens of different sizes but interpret their findings

then in terms of size and flexibility of the constructs. There is no clear distinction being made between these two physical parameters, see e.g. Figure 8 which states "size and flexibility", "too large and flexible", "too small and rigid" on a single axis. This pairing of properties is correct for the constructs themselves but obscures the proposed concept. Have the authors considered that there might be an optimal antigen footprint (=size) itself for activation (without the need for flexibility)? Would it be possible to distinguish between these two properties by introducing mismatches into the "optimal" 8-7-8 scaffold to make it more flexible without affecting its size? Moreover, there is no suggestion provided in the discussion how mechanistically the size (and/or flexibility) of the antigen could affect BCR activation. Do the authors envision that binding of the macromolecular antigens exerts a mechanical strain on the binding site of the receptor, eventually leading to conformational changes (see ref 57)? To resolve these issues, either the discussion and title require rewording/focussing, and/or additional experiments would be needed.

Line 325f: "[...] activation was not a simple function of degree of 'loading' of the cells with HJ (Fig. 4D and E)." For making this point clearer in the results text, the case of the HJ-3xNIP could be described which showed maximal activation at 75:1 (Fig. 4C) but only intermediate binding (Fig. 4E).

Reviewer #4 (Remarks to the Author):

This is a novel study that utilizes novel experimental and computational approaches to investigate key aspects of antigen footprint. The reviewers have thoroughly addressed all the points that were raised by the original reviewer, and therefore should be accepted for publication.

We would like again to begin by thanking the Reviewers for their careful and constructive evaluation of our work. Below we provide a point-by-point response to the points raised. The main changes to the manuscript can be summarized as follows:

- As requested by Reviewer 1, we have provided an argument for the physiological relevance of the antigen doses employed in our *in vitro* experiments, in relation to those of commonly used immunization strategies *in vivo*, as well as an infection scenario in humans. We have included a concise version of the former argument in the Discussion section of the revised manuscript.
- Furthermore, based on the Reviewers' recommendations, we have revised Figures 4 and 7, using an easier distinguishable color scheme for Figure 4, panels B-E, and Figure 7, panel B, and including the base structure of the Q4-1NIP structure in Figure 7, panel A, for consistency.
- We have additionally revised the graphical summary of our findings presented in Figure 8, in line with the suggestions of Reviewers 1 and 3.
- To comply with journal policies, we have updated the flow cytometry plots in Figures 3, S9 and S10 from contour plots to contour plots with outliers.
- Furthermore, we have represented internalization data in Figure 5 with 95% CI and included statistical significance of comparisons.
- Minor changes have been made in the text, in line with the above, and to clarify additional points raised in the review.
- We have also streamlined the parts of the Results section covering the binding of the HJ-Ag structures to the antibody and cells, as suggested by Reviewer 2.
- Finally, on re-checking all our calculations and presented numbers, we discovered a small rounding error on the third decimal for the number of nanobodies bound per BCR given in the Results section concerning the DNA-PAINT analyses. The labeling density and all other listed values were correct. The correct number of nanobodies bound per BCR was 1.545, which should have been rounded to 1.55 or 1.5, but by mistake was listed as 1.6. We apologize for this error, which has now been corrected in the text.

REVIEWER COMMENTS

Reviewer #1 (Remarks to the Author):

The authors have responded largely my concerns and the manuscript has been improved. I only have the following remaining comments.

Regarding the response to comment 1- I still doubt that the recalculated molar ratio is anywhere close to physiological. The authors argued that no one really known what the antigen availability is *in vivo*. But there is good information about the antigen availability for TCR *in vivo*. Do the authors really think there are such high excess of antigen compared to BCR?

We fully acknowledge the necessity of substantiating the physiological relevance of the employed antigen doses. Based on common antigen doses used in immunization

experiments, we can provide a rough estimate of this. For immunization experiments in mice, it is common to administer 100 ng – 1 µg antigen in the hock or subcutaneously in the flank, for example. Commonly used hapten-protein carrier conjugates include NP- or NIP-hen egg lysozyme (HEL) or ovalbumin (Ova), with respective molecular weights of roughly 15 and 45 kDa, respectively. Antigen would drain into the popliteal or inguinal lymph node, by the two administration routes, respectively, and these would contain around 1.5-3 million lymphocytes (regular C57BL/6). Approximately 40% of these would be B cells, and depending on the setting, typically 1-10% of these could be antigen-specific (depending on whether adoptive transfers of B1-8 cells are used, etc.). Assuming efficient drainage and distribution in the lymph node, and using our determined BCR density, we can calculate the lower and upper limits of antigen:BCR ratios *in vivo* as:

NP-/NIP-Ova (45 kDa), 100 ng dose, draining to inguinal LN containing 3 million lymphocytes, with 40% B cells of which 10% Ag-specific:

$$\frac{100 * 10^{-9} \text{ g}}{45,000 \frac{\text{g}}{\text{mol}}} * 6.022 * 10^{23} \frac{\text{molecules}}{\text{mol}} = 1.34 * 10^{12} \text{ molecules antigen}$$

$$3 * 10^6 \text{ lymphocytes} * 40\% \text{ B cells} * 10\% \text{ Agspecific} * 25,000 \frac{\text{BCRs}}{\text{cell}} = 3 * 10^9 \text{ BCRs}$$

Ratio of Antigen:BCR = 446 : 1

NP-/NIP-HEL (15 kDa), 1 µg dose, draining to popliteal LN containing 1.5 million lymphocytes, with 40% B cells of which 1% Ag-specific:

$$\frac{1 * 10^{-6} \text{ g}}{15,000 \frac{\text{g}}{\text{mol}}} * 6.022 * 10^{23} \frac{\text{molecules}}{\text{mol}} = 4.01 * 10^{13} \text{ molecules antigen}$$

$$1.5 * 10^6 \text{ lymphocytes} * 40\% \text{ B cells} * 1\% \text{ Agspecific} * 25,000 \frac{\text{BCRs}}{\text{cell}} = 1.5 * 10^8 \text{ BCRs}$$

Ratio of Antigen:BCR = 267,334 : 1

Several factors will modify the actual antigen availability *in vivo* of course. Likely, the distribution volume *in vivo* is larger, yielding a lower concentration, despite the comparable ‘antigen to BCR ratio’. Furthermore, often (but not always) antigen is administered in e.g., alum or mineral oil, causing a depot effect with slower gradual release and hence lower relative concentration at a given time point. Finally, some antigen may be degraded or cleared by subcapsular sinus macrophages etc. On the other hand, most commercially available or researcher-generated hapten-carrier reagents have multiple haptens conjugated per carrier molecule, i.e., NP₅-HEL or NP₁₅-Ova (used by us as a control in the present manuscript), which

would yield a higher effective ratio of antigen to BCR. Furthermore, antigen will typically be transported to and deposited on follicular dendritic cells in the B cell follicles, causing a local up-concentration of antigen.

We believe the above calculations and considerations substantiate that our antigen to BCR ratios in the presented experiments are in fact commensurate with what is encountered in an immunization scenario, and hence relevant to physiological conditions. We have included a brief version of this consideration in the Discussion.

We can further extend such considerations to a natural infection scenario in humans. It was recently estimated that each infected person carries 1 billion to 100 billion virions during peak infection (<https://www.pnas.org/doi/10.1073/pnas.2024815118>). Each virion carries around 20-40 spike proteins (<https://www.science.org/doi/10.1126/science.abd5223>). The median frequency of receptor-binding motif (RBM)-specific B cells among total B cells in 8 seronegative donors was determined to be 0.0025% (<https://www.science.org/doi/10.1126/sciimmunol.abl5842>). With 10 billion B cells in the human body, this would amount to 250,000 RBM-specific B cells, each carrying 25,000 bivalent receptors:

Spike antigen: 1-100 billion virions * 20-40 spike proteins/virion = 20-4,000 billion molecules

RBM-specific B cell Fabs: 250,000 RBM-specific B cells * 25,000 BCR per cell * 2 Fab arms/BCR = 12.5 billion Fabs

Ratio of Antigen:BCR = 1.6:1 to 320:1

Again, numerous factors could impact this number, such as multiple overlapping or non-overlapping target epitopes on the antigen, shedding of spike protein by the virus and virus-infected cells, competition with secreted antibodies, distribution and bio-availability of viral particles/shedded antigens, and sub-availability of the B cell repertoire.

This antigen availability is lower compared to the immunization scenario, and indeed, it seems probable that in a vaccination setting, larger antigen doses are encountered than what would be presented in the context of an infection (at least initially). Still, the antigen ratios appear to be overall well in agreement with the ratios employed in our *in vitro* setting.

Regarding the response to comment 4- The authors updated Figure 4 and added Figure S14 to show that NP-FITC with various molar excess could not activate properly B cell signaling. I appreciate that the relationship between size, flexibility and BCR activation activity is complex. Can the authors show this in a quantitative way using the data in Fig 7? For example, plots of Activation (fold-change of Ca²⁺ signaling) vs antigen size (gyro radius) and flexibility (order of rotational symmetry) would help for readers to visualize 'quantitative footprint-requirement of antigen to properly activate B cell'.

We acknowledge the point and thank the Reviewer for the suggestion to directly couple the data on activation potential with the size and flexibility of the antigen. Unfortunately, it is not

possible to quantitatively model the gyro radii and flexibility of the dumbbell structures, as we cannot predict the preferred 3D conformation of the constructs. We can only relatively estimate that one antigenic structure is larger than another, or that the increase in size of one structure is likely offset by a concomitant increase in flexibility. Perhaps more importantly, we have not seen any indications that there is such a thing as an optimal size for activation, quite the contrary, we now have data using a DNA nanopore structure (in this case not acting as a pore, but as a massive, rigid 22x32 nm structure generated by DNA origami) to dramatically increase the size of our hapten-conjugated HJ construct, which demonstrates that ‘bigger is better’ in terms of activation potential of monovalent antigen. However, it is unlikely that such a sizeable monovalent antigen would ever be encountered in nature, where larger antigenic entities (intact viruses or bacteria) typically present arrays of antigens. In essence, we think that BCR recognition is working on two different scale levels to recognize smaller, often monovalent antigens (e.g., bacterial toxins) and larger often multivalent antigens (again, viruses and bacteria), whereas innocuous small-molecular antigen is ignored. Here we have focused on the discrimination of small-molecular and small mono- or poly-valent antigens, and we feel it is beyond the scope of the present manuscript to include our DNA nanopore data, but we are working on further delineating the exact requirements for the footprint and the mechanism whereby this translates into activation potential.

We realized though, also prompted by a similar critique by Reviewer 3, that Figure 8 did not appropriately represent the size vs. flexibility aspect. It is inherently challenging to represent the relationship between two different parameters on a single axis. The labeling of the x-axis in the right panel of Figure 8 has been corrected to “Size / Flexibility”, to indicate that increasing size increases activation, whereas increasing flexibility decreases activation. Furthermore, “Too small and rigid” has been corrected to “Too small” (as rigidity is required for efficient activation), and “Too large and flexible” has been corrected to “Too flexible” (as increasing size in itself increases the activation).

We have additionally clarified our phrasing regarding what can be concluded from the presented observations in the manuscript text:

”Surprisingly, there was not a linear relationship between the size of the scaffold and its activity. The 8-7-8-Q4-1NIP dumbbell structure elicited the most potent response, suggesting that differences in the flexibility or preferred 3D configuration of the various dumbbells may also influence their activity. Our overall findings regarding affinity- and avidity-dependent activation and the role of antigen footprint in monovalent antigen-driven activation are illustrated schematically in **Figure 8.**”

Please also see to our additional considerations regarding the mechanism of activation in our response to Reviewer 3 below.

Reviewer #2 (Remarks to the Author):

The new manuscript is greatly improved and generally addresses my comments. I also enjoyed the clearer discussion. I only have a few minor issues:

1. The description of the binding of the HJ-Ag structures to the antibody and cells (pages 9-15) may be condensed as it only serves as a control and does not produce any novel information.

In the revised manuscript, we have tightened the text in the sections describing the binding of the HJ-Ag structures to the antibody and cells.

2. Line 326 is unclear. At this point, the data still show a correlation between binding and activation, because I don't see much binding detected for the monovalent HJ at concentrations that do not activate calcium flux (Fig 4E). This point becomes clearer later with the low-MW antigens. Perhaps lack of clarity also comes from the next issue.

We apologize for the lack of clarity here. For example, HJ-3xNIP at the high stoichiometry yields a binding signal comparable to that of HJ-1xNP (Figure 4D), but elicits a much more robust activation (Figure 4B), and although HJ-2xNP binds robustly at the low stoichiometry (Figure 4E), it yields only a weak activation signal (Figure 4C). However, we see how the color scheme used to delineate the different conditions was less than ideal, and this has been corrected. Furthermore, we have added the former example in-text, as also suggested by Reviewer 3:

"Moreover, we could experimentally verify 'in-assay' that at the low stoichiometry, HJ-1xNIP did indeed bind the B cells, and in general, activation was not a simple function of degree of 'loading' of the cells with HJ (Fig. 4D and E). For example, whereas HJ-3xNIP at high stoichiometry only yielded a binding signal comparable to that of HJ-1xNP (Figure 4D), it elicited a much more robust activation (Figure 4B)."

3. Fig 4E and Fig 7 would benefit from different colors of binding/calcium lines. It is very difficult to distinguish the shades of grey or orange.

We have corrected the color scheme to allow easier distinction. For clarity of the employed structures, we additionally also included Q4-1NIP in the schematic representation in Fig. 7A.

Reviewer #3 (Remarks to the Author):

The revised manuscript by Ferapontov et al. includes new data to elucidate the concept of the antigen footprint, as well as several control experiments and changes to the manuscript text to clarify technical questions raised by all reviewers. This revision constitutes a significant improvement, however, minor issues remain which, if adequately addressed, could further support the data interpretation.

Regarding Point 1.b raised during my original review: thank you for clarifying that only the lower membrane is being imaged. The confusion arose from the xyz 3D view in Figure 1 B,

lower panel, which shows a difference in height (z) of ca 700 nm, while the length of the line profile is $\sim 6 \mu\text{m}$. The inner $\sim 4 \mu\text{m}$ show signals at the bottom (blue) but also higher up (yellow and red) – what are these latter signals? Are these BCRs on membrane ruffles? A better description of this Figure in the results section would be helpful to avoid confusion of the readers. If these are membrane ruffles, are these (fully) contained in the imaging depth of the qPAINT measurements? In that case it would technically speaking not be correct to use a correction factor of 2 for the extrapolation of the BCR count per cells.

We apologize for the lack of clarity in our initial response. The image in Figure 1B lower panel is derived from 3D imaging. The signals referred to by the Reviewer could stem from membrane ruffles, as suggested, or alternatively from BCRs located within the cell (e.g., in ER, Golgi, or endosomal vesicles). However, for qPAINT analyses we exclusively performed 2D TIRF imaging, within the $\sim 100 \text{ nm}$ evanescent field proximal to the cover slip, effectively excluding this ‘deeper’ signal observed upon 3D imaging. We realize that this was not entirely clear from our initial response and apologize for the lack of precision. We have additionally clarified this point in-text:

“For image analysis, we used 2D TIRF images with an imaging depth of approximately 100 nm, and selected either a rectangular or a round region of interest, covering the majority of the lymphocyte touching the surface but excluding the edges (Fig. 1B).”

The concept of antigen footprint still remains vague. The authors have performed additional experiments with different dumbbell shaped antigens of different sizes but interpret their findings then in terms of size and flexibility of the constructs. There is no clear distinction being made between these two physical parameters, see e.g. Figure 8 which states “size and flexibility”, “too large and flexible”, “too small and rigid” on a single axis. This pairing of properties is correct for the constructs themselves but obscures the proposed concept. Have the authors considered that there might be an optimal antigen footprint (=size) itself for activation (without the need for flexibility)? Would it be possible to distinguish between these two properties by introducing mismatches into the “optimal” 8-7-8 scaffold to make it more flexible without affecting its size? Moreover, there is no suggestion provided in the discussion how mechanistically the size (and/or flexibility) of the antigen could affect BCR activation. Do the authors envision that binding of the macromolecular antigens exerts a mechanical strain on the binding site of the receptor, eventually leading to conformational changes (see ref 57)? To resolve these issues, either the discussion and title require rewording/focussing, and/or additional experiments would be needed.

We acknowledge the Reviewer’s point. It is inherently challenging to represent the relationship between two different parameters on a single axis. The labeling of the x-axis in the right panel of Figure 8 has been corrected to “Size / Flexibility”, to indicate that increasing size increases activation, whereas increasing flexibility decreases activation. Furthermore, “Too small and rigid” has been corrected to “Too small” (as rigidity is required for efficient activation), and “Too large and flexible” has been corrected to “Too flexible” (as increasing size in itself increases the activation). We have not seen any indications that there is such a thing as an optimal size for activation, quite the contrary, we now have data using a DNA nanopore structure (in this case not acting as a pore, but as a massive, rigid 22x32 nm

structure generated by DNA origami) to dramatically increase the size of our hapten-conjugated HJ construct, which demonstrates that bigger is in fact better in terms of activation potential of monovalent antigen (please also see our response to Reviewer 2's second point). We think it is beyond the scope of the present manuscript to include this data, but we are working on further delineating the exact requirements for the footprint and the mechanistic link to activation potential.

We have clarified our phrasing regarding what can be concluded from the presented observations in the manuscript text:

"Surprisingly, there was not a linear relationship between the size of the scaffold and its activity. The 8-7-8-Q4-1NIP dumbbell structure elicited the most potent response, suggesting that differences in the flexibility or preferred 3D configuration of the various dumbbells may also influence their activity. Our overall findings regarding affinity- and avidity-dependent activation and the role of antigen footprint in monovalent antigen-driven activation are illustrated schematically in **Figure 8**."

We do not envision that binding of the macromolecular antigens exerts a mechanical strain on the binding site of the receptor, because there is no physical framework for such an 'induced fit' mechanism of action. The short-range binding interactions of individual ligand-receptor pairs and the involved free energy would be unable to drive the required change in conformation. Rather, we envision that docking a bulky and rigid structure to BCR(s) on the surface of the B cell will perturb the dynamic equilibrium movement of CD45 into the contact zone, causing a spatial depletion of CD45 from the vicinity of the engaged BCR(s), thereby shifting the balance of kinases and phosphatases in favor of the former. Antigen factors favoring activation would include multivalency (engagement of multiple BCRs), size (larger exclusion zone), and potentially charge (negative charge would increase repulsion of CD45, but this is again beyond the scope of the present work), whereas increased flexibility would disfavor activation (reduction of the exclusion zone). We have attempted to clarify this in the discussion:

"Our further definition of the antigen footprint using DNA dumbbell nanostructures confirm size as a critical parameter, but reveal that rigidity likely plays an additional role, in line with the suggested requirement for the antigen to displace CD45 from the vicinity of the BCR in order to elicit activation. We envision that docking a bulky and rigid structure to BCR(s) on the surface of the B cell will perturb the dynamic equilibrium movement of CD45 into the contact zone, causing a spatial depletion of CD45 from the vicinity of the engaged BCR(s), thereby shifting the balance of kinases and phosphatases in favor of the former. Antigen factors favoring activation would include multivalency (engagement of multiple BCRs), size (larger exclusion zone), and potentially charge (negative charge could increase repulsion of CD45), whereas increased flexibility would disfavor activation (reduction of the exclusion zone)."

Line 325f: "[...] activation was not a simple function of degree of 'loading' of the cells with HJ (Fig. 4D and E)." For making this point clearer in the results text, the case of the HJ-3xNIP could be described which showed maximal activation at 75:1 (Fig. 4C) but only intermediate binding (Fig. 4E).

We thank the Reviewer for this excellent suggestion. We have added this example in the in-text description, and additionally changed the color scheme for increased clarity as suggested by Reviewer 2:

”Moreover, we could experimentally verify ‘in-assay’ that at the low stoichiometry, HJ-1xNIP did indeed bind the B cells, and in general, activation was not a simple function of degree of ‘loading’ of the cells with HJ (**Fig. 4D and E**). For example, whereas HJ-3xNIP at high stoichiometry only yielded a binding signal comparable to that of HJ-1xNP (**Figure 4D**), it elicited a much more robust activation (**Figure 4B**).”

Reviewer #4 (Remarks to the Author):

This is a novel study that utilizes novel experimental and computational approaches to investigate key aspects of antigen footprint. The reviewers have thoroughly addressed all the points that were raised by the original reviewer, and therefore should be accepted for publication.

We thank the Reviewer for the positive reception, and for contributing an additional evaluation of our manuscript and initial rebuttal.

We would like to once again express our gratitude to all the Reviewers for their careful, critical and constructive evaluation of our work, which we believe has significantly strengthened the manuscript.

REVIEWERS' COMMENTS

Reviewer #1 (Remarks to the Author):

The authors have responded to my comments satisfactorily. I have no more concerns.

Reviewer #2 (Remarks to the Author):

My comments have been addressed and I support the publication of this improved manuscript.

Reviewer #3 (Remarks to the Author):

The authors have responded to my previous queries in a careful and convincing manner. The amended manuscript has gained clarity and the results are discussed fairly. Especially the concept of the antigen footprint has been worked out better, and is an interesting new concept in B-cell activation.